# Multimodal locomotion ultra-thin soft robots for exploration of narrow spaces

Xi Wang [1], Siqian Li[1], Jung-Che Chang[1], Jing Liu [1], Dragos Axinte[1] & Xin Dong [1] ✉

From power plants on land to bridges over the sea, safety-critical built environments require periodic inspections for detecting issues to avoid functional discontinuities of these installations. However, navigation paths in these environments are usually challenging as they often contain difficult-to-access spaces (near-millimetre and submillimetre-high gaps) and multiple domains (solid, liquid and even aerial). In this paper, we address these challenges by developing a class of Thin Soft Robots (TS-Robot: thickness, 1.7 mm) that can access narrow spaces and perform cross-domain multimodal locomotion. We adopted a dual-actuation sandwich structure with a tuneable Poisson's ratio tensioning mechanism for developing the TS-Robots driven by dielectric elastomers, providing them with two types of gaits (linear and undulating), remarkable output force (~ 41 times their weight) and speed (1.16 times Body Length/s and 13.06 times Body Thickness/s). Here, we demonstrated that TS-Robots can crawl, climb, swim and collaborate for transitioning between domains in environments with narrow entries.

Locomotion in built environments is a great challenge for all robots, large and small, as various types of obstacles, such as doors, stairs and narrow gaps, are often present in the navigation paths preventing them from exploring the targets. To address this challenge, numerous great robots have been developed[1], including humanoid[2], wheel/track-based[3] and quadruped types[4,5], which can move on different terrains and even replicate some human abilities (e.g. open doors[6]) for overcoming macro-scale obstacles. In contrast, for accessing small and confined spaces, a family of small robots, such as worm[7–10] and insect-like configurations[11–14] have been proposed. Generally, each of these is designed with one locomotion mode for crossing over a specific type of obstacle. However, when it comes to many real-world applications, e.g., inspections of high-value assets (e.g. power plants), complex built spaces (e.g. industrial plants) and civil engineering dwellings (e.g. bridges), these require robots with multiple locomotion modes to advance through spaces containing multiple types of obstacles and even multiple domains (solid, liquid and even aerial), which remains a challenge for the robotics community.

In civil/industrial built environments, narrow accesses (at millimetre scales) commonly occur, such as spaces under doors or heavy objects (e.g., containers), duct systems for air passages/circulation, and gaps within complex machinery (e.g., gas turbine engines). This requires robots to be designed with millimetre-scale thickness[15,16] and more importantly, with multimodal locomotion capabilities for passing through narrow gaps and other types of obstacles commonly seen in these environments. To achieve this goal, many robotic systems, in particular soft robots, have been developed to achieve locomotion by either in-plane compression/extension or out-of-plane bend, driven by pneumatics[17,18], dielectric elastomers[19,20], shape memory alloys[7], thermally sensitive polymers[21] or by using external magnetic fields[9,22]. Among these soft robots, most small-scale systems are developed for medical surgery or drug delivery within human bodies; these are usually actuated by external magnetic fields[23,24], of which a few perform multimodal locomotion in narrow spaces and transition between multiple domains (solid and liquid). However, within human-made environments, this highly versatile magnetic field manipulation could prove difficult to adapt to control the robots, due to the large

[1]Rolls-Royce University Technology Centre in Manufacturing and On-Wing Technology, Faculty of Engineering, University of Nottingham, NG7 2GX Nottingham, UK. ✉e-mail: Xin.Dong@nottingham.ac.uk

dimensions of these environments and the inherent interferences from commonly used ferromagnetic materials, such as steel reinforcement bars and pipes.

Apart from the magnetic-driven approach, dielectric elastomer actuators (DEAs) have been widely used for building small mobile robots, owing to their outstanding features of high-power density, flexibility and robustness[25]. Recent studies have demonstrated several designs of miniature DEA-driven soft robots are capable of performing single or multiple locomotion modes within single domains, such as jumping and crawling, for environmental monitoring and deployment in challenging workspaces[19,26,27], thus, proving DEAs as a suitable option for building thin multiple-DoFs robots. Furthermore, locomotion in built environments often requires robots to climb on inclined or even vertical surfaces, such as slopes and walls. This requires a controlled approach to attach the robots to these surfaces reversibly. A variety of approaches have been undertaken to achieve miniature robot attachment to vertical surfaces, such as bio-inspired dry adhesion[28,29], vacuums[17], and electroadhesion[26,30,31], with the latter solution miniaturised for the design of light/thin robots.

In this paper, we propose a new approach to develop Thin Soft Robots (TS-Robots) capable of performing multimodal locomotion in both solid and liquid domains, including the ability to transition between them to explore difficult-to-access human-made environments. The design of the TS-Robots consists of a Thin Soft Dielectric Elastomer Actuator (TS-DEA; thickness, 1.2 mm) and multiple Electrostatic Adhesive Pads (EA-Pads), which can be assembled in symmetric and asymmetric forms. Furthermore, the capabilities of our TS-Robots can be enhanced by joining them together, enabling navigation within complex environments while acting as manipulators within confined spaces. The TS-Robots can perform as a carrier for delivering goods or other types of robots (e.g., drones), enabling the "carrier" and "passenger" with additional locomotion modes. In this paper, we demonstrated the concept of single/joined TS-Robots for multimodal locomotion by performing independent and collaborative crawling, climbing, steering, swimming, landing, and passive flying, as well as transitions between surfaces and programmable complex manipulations.

## Results

### Design of the TS-Robots for cross-domain multimodal locomotion

The design of the TS-Robots consists of two sub-systems, including a Thin Soft Dielectric Elastomer Actuator (TS-DEA) for generating displacement and driving forces, and multiple Electrostatic Adhesive Pads (EA-Pad) for anchoring the robot to the substrate surfaces (Fig. 1).

We used a dual-actuation sandwiched structure for designing the TS-DEA, which contains two actuation layers driving a compressible tensioning mechanism in the middle of them (Fig. 1a). Each actuation layer consists of a flexible elastomer, sandwiched between two electrodes, which is bonded to the tensioning mechanism through an adhesive layer. Based on this design, the TS-DEA can perform two motion types: one in-plane linear compression/extension by synchronising the displacement of the two actuation layers and one out-of-plane bending motion by differencing the displacement of both. Depending on the requirements of the applications, both VHB-based and silicone-based elastomers can be selected and used in the actuation layer. Further, more importantly, this design enables us to easily tune the resonant frequency of the TS-DEA for fast actuation by mechanically changing the properties (e.g., stiffness) of the tensioning mechanism, which could be less time-consuming to process compared with the widely used state-of-the-art chemical method[32] (Supplementary file Text S1).

Further, we applied a design of negative or zero Poisson's ratio to the compressible tensioning mechanism, serving as the skeleton for stretching the actuation layers and tuning its displacement ratio in two orthogonal directions (X and Y). When a stretching force is applied, a negative Poisson's ratio tensioning mechanism presents a coupled extension motion in two orthogonal directions (along and perpendicular to the force), while a zero Poisson's ratio mechanism provides an elongation movement in the direction of the force, but without contraction in the orthogonal direction. Such mechanisms are also known as re-entrant structures[33], and their Poisson's ratio ($v$), as well as its displacement ratio between the X and Y directions, can be calculated by

$$v = -\frac{\sin\theta \, x_i}{\cos\theta \, y_i} \tag{1}$$

where $v$ is the Poisson's ratio of the mechanism, $\theta$ represents the angle of the diagonal strut with respect to Y-axis, $x_i$ and $y_i$ are the width and length of a single re-entrant cell (Fig. 1b and Supplementary Fig. S3). When the values of $h$ and $l$ are given, the Poisson's ratio of the tensioning mechanism can be tuneable by adjusting the angle of the diagonal strut $\theta$ (Fig. 1b, see details in the Supplementary file Text S2). Consequently, the TS-DEA can be designed with anisotropy motion along X and Y axes by tuning the Poisson's ratio of the tensioning mechanism.

To describe the linear motion of the TS-DEA with respect to its input voltage, we developed an analytical model that considers the force generated by the actuation layers and the tensioning mechanism and the interaction between both (Fig. 1c).

$$A(\sigma_1(\Delta) + \sigma_2(\Delta)) = (F_{M1}(U,\Delta) + F_{M2}(U,\Delta)) + F_{\text{Re}}(\Delta) \tag{2}$$

where $A$ is the area of cross-section of the actuation layers perpendicular to the extension direction; $\sigma_1(\Delta)$ and $\sigma_2(\Delta)$ are the stress of the two actuation layers at the TS-DEA displacement $\Delta$[34–36]; $F_{M1}(U,\Delta)$ and $F_{M2}(U,\Delta)$ represent the voltage-induced extension force of the actuation layers along the extension direction, which is analysed by using thermodynamic theory[37]; $U$ is the input voltage of the actuation layers; $F_{\text{Re}}(\Delta)$ is the extension force[38,39] generated by the tensioning mechanism at $\Delta$ (see details in the Supplementary file Test S3 and Text S5).

Further, the bending motion of the actuator is achieved by actuating one of the actuation layers and its model can be written as

$$F_{\text{Re}}(u) = A(\sigma_1(u) + \sigma_2(u)) - (F_{M1}(U,u) + F_{M2}(U,u)) \tag{3}$$

$$M_{\text{Re}}(u) = ((F_{M2}(U,u) - F_{M1}(U,u)) + A(\sigma_1(u) - \sigma_2(u)))\frac{t_T}{2} \tag{4}$$

where $M_{\text{Re}}(u)$ is the bending moment of the TS-DEA at the tip deflection of $u$; $t_T$ is the thickness of the tensioning mechanism and we assume the thickness of the actuation layer is negligible (Fig. 1d, see details in the Supplementary file Test S4 and S5).

As enabling elements of the TS-Robots, the EA-Pads are designed using well-established electrostatic adhesion technology[26,40] to generate the attraction force with substrate surfaces during the locomotion in solid domains. The design and fabrication process of the EA-Pads is presented in the Supplementary file Text S6.

As such, in this paper, we fabricated the following three families of robots as samples for evaluating the performance of the thin-soft robots that are able for cross-domain multimodal locomotion:

- Type-A TS-Robot: A design composed of a zero Poisson's ratio TS-DEA ($v = 0$, denoted as Type-A TS-DEA, Fig. 1e) with two EA-Pads; the robot can perform extension/contraction and bending motions in one direction.
- Type-B TS-Robot: A system empowered by a negative Poisson's ratio TS-DEA ($v = -1$, denoted as Type-B TS-DEA, Fig. 1f) with four EA-Pads; the robot can conduct extension/contraction to steer in two directions.

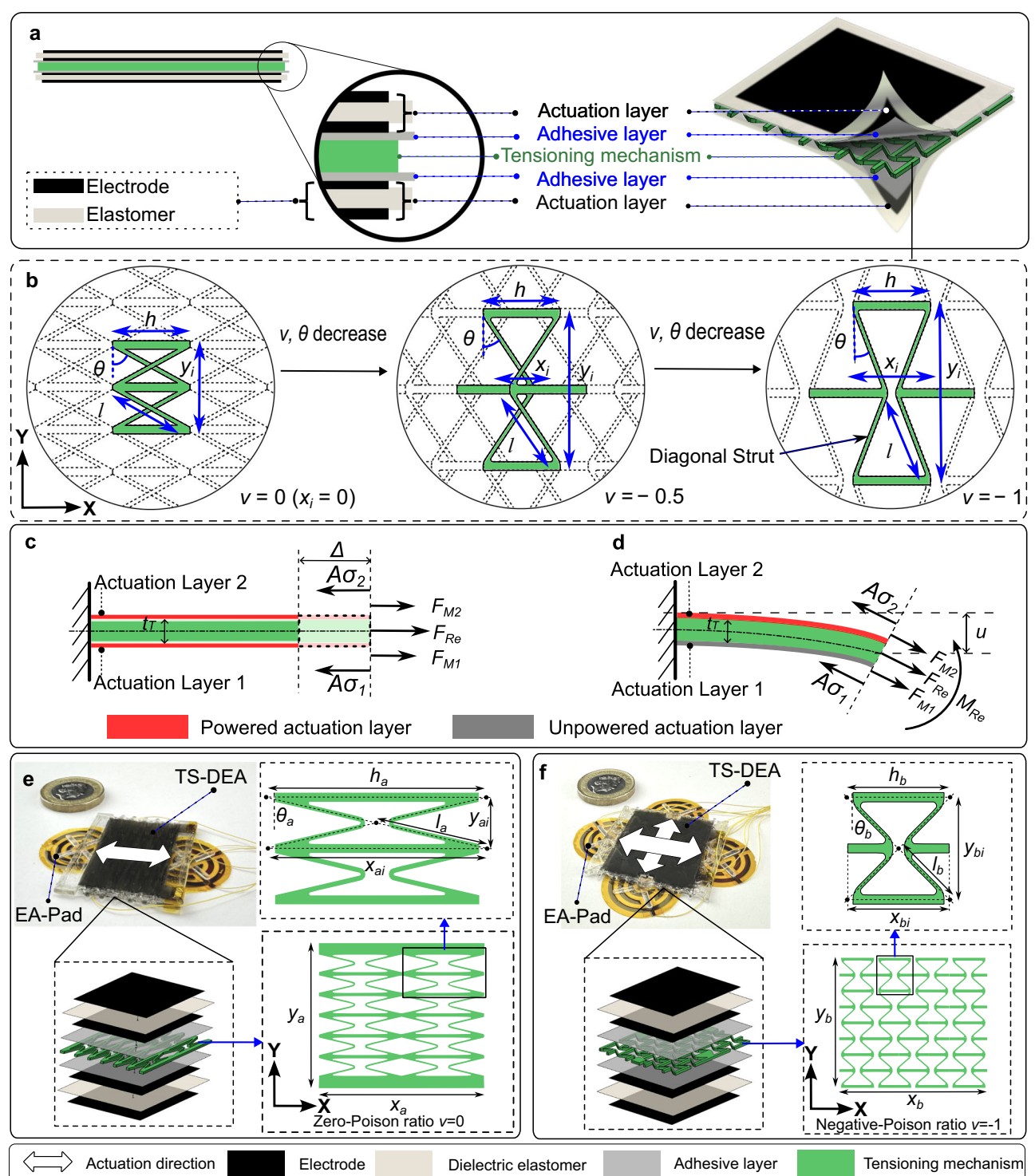

**Fig. 1 | Design of thin soft dielectric elastomer actuators (TS-DEAs). a** The dual-actuation sandwich structure of the TS-DEAs; **b** Design of the tensioning mechanisms with tuneable Poisson's ratio (in the present case, $h$ and $l$ are kept constant while $\theta$ is variable); **c** Working principle and force diagram of the TS-DEAs for in-plane linear motion; **d** Working principle and force diagram of the TS-DEAs for out-of-plane bending motion; **e** Design of the Type-A TS-Robot (the Poisson's ratio of its tensioning mechanism is zero; One directional motion); **f** Design of the Type-B TS-Robot (the Poisson's ratio of its tensioning mechanism is −1; Two directional motions).

Here, the EA-Pads offer a great advance, enabling the proposed robots to move on the surfaces of various materials and orientations. However, as the polarisation and depolarisation time of the substrate surfaces by the EA-Pads is 100-300 ms per cycle, this bottleneck prevents the proposed TS-Robot from being operated with a high frequency. Hence, we used VHB as the elastomer in the actuation layers

for Type-A and -B TS-Robots since it can withstand high electrical field strength to generate large strain and force at low frequencies (1-5 Hz).

As a complement, we also tested other designs of TS-Robots for fast actuation by using two types of silicone-based elastomers[32] (one is a mixture material composed of Ecoflex 00-30 and Sylgard 184 in a 1:1 ratio, in which Sylgard 184 was prepared with a cross-linker ratio of

40:1; the other one is made of pure Ecoflex 0030) and directional friction foot instead of VHB and EA-Pads, respectively.

- Type-C TS-Robot: the system contains a zero Poisson's ratio TS-DEA with directional friction foot. Its actuator is denoted as Type-C TS-DEA, with its actuation layers made of a mixture of materials (Supplementary file Text S1).
  - I. Type-C-I TS-DEA: the actuator is made of a zero Poisson's ratio tensioning mechanism (stiffness $k_T$) sandwiched by two actuation layers (the mixture material).
  - II. Type-C-II TS-DEA: the actuator comprises a zero Poisson's ratio tensioning mechanism (stiffness $2k_T$), sandwiched by four actuation layers (the mixture material), two on each side.
  - III. Low-Voltage-Type-C TS-DEA (L-Type-C TS-Robot): the actuator comprises a zero Poisson ratio tensioning mechanism, sandwiched by four actuation layers (pure Ecoflex 0030; two on each side; each layer is 10um in thickness), for low-voltage actuation.

## Characterisations of the TS-DEAs

The capability of the TS-Robots was analysed by investigating the characteristics of two sub-systems, the TS-DEAs and EA-pads, respectively. We characterised the static and dynamic properties (Supplementary Movie 1 and 2) of the Type-A, Type-B and Type-C TS-DEAs, including their displacement and output force, using the experimental setup presented in Supplementary Fig. S22. As a well-established approach to developing the EA-pads was applied, we present its characterisation summarised in Supplementary file Text S7.

As shown in Fig. 2a and b, the Type-A TS-DEA has a maximum displacement of 2.39 mm (6.0% strain, 100 V/um) and a maximum static blocking force of 0.57 N (50um thickness, 100 V/um) when constrained to its initial length. Although both actuation layers are composed of a single-layer elastomer (VHB4910, 50um thickness after stretch), the Type-A TS-DEA still achieve a high force/weight ratio, where the maximum blocking force is 48 times the weight of the TS-DEA (11.7mN) and 291 times that of two actuation layers (1.96mN). We investigated the dynamic properties of the Type-A TS-DEA, actuated at the electric field strength of 100 V/μm by a square wave linear frequency sweep ranging from 1 Hz to 100 Hz. Due to the viscoelastic properties of VHB, the displacement of the Type-A TS-DEA decreased to below 0.1 mm when frequencies reached 20 Hz (Fig. 2c). Hence, considering the polarisation and depolarisation time of the substrates by the EA-Pad (less than 10 Hz), the demonstrations of the Type-A TS-Robot were primarily conducted at lower frequencies, specifically within the 1-5 Hz range.

Further, we characterised the static and dynamic properties of the bending motion of the Type-A TS-DEA. The maximum static bending displacement of the TS-DEA is 4 mm at the voltage of 5 kV, which is 10% of the body length (Fig. 2d); the maximum blocking force of the actuator is 17.5mN, measured from 40 mm away from its fixed end (Fig. 2e), leading to a maximum torque of 0.7mNm. Figure 2f illustrates the experimental sweep frequency of the actuator's displacement at 5 kV from 1 to 50 Hz. The resonate frequency of the TS-DEA is at 12 Hz with a displacement amplitude of 14 mm (35% of the body length), nearly four times as large as the maximum static bending displacement.

Similarly, we characterised the static properties of the Type-B TS-DEA, in terms of displacement and force (Fig. 2g and h). The maximum displacement of the Type-B TS-DEA in the X and Y direction are 0.65 and 0.67 mm (1.6% and 1.7% strain), respectively. These results align with the predicted performance of the negative Poisson's ratio ($\nu = -1$) design of the tensioning mechanism (the displacement in $X$ axis is equal to that in $Y$ axis). Further, the maximum blocking force of the Type-B TS-DEA is 0.51 N (43 times the weight of the actuator; 260 times the weight of the two actuation layers) on the $X$-axis and 0.52 N (44 times the weight of the actuator; 265 times the weight of the two actuation layers) on the $Y$-axis. The negligible difference in

displacement and force in the X and Y axes may come from the manufacturing and assembly tolerance of the actuators. It can be found that the dynamic displacements of the Type-B TS-DEA in both X and Y directions decrease with the rise of the driving frequency (Fig. 2i), and the magnitude of the displacement in the two directions are different, as the mass distribution of the tensioning mechanism in X and Y axes are distinctive.

Further, it was found that both Type-C-I and -II DEAs can make a maximum static displacement of 0.9 mm (Fig. 2j) and a blocking force of 0.038 N and 0.081 N (Fig. 2k), respectively. Then, the dynamics of Type-C-I DEA (tensioning mechanism stiffness $k_T$, driven by two actuation layers) were tested at an electric field strength of 21.7 V/um. We found that its resonant frequency can reach 51 Hz, and its displacement is 0.22 mm. In contrast, the resonant frequency of Type-C-II DEA (tensioning mechanism stiffness $2k_T$, driven by four actuation layers) is 67 Hz, and its displacement is 0.13 mm (Fig. 2l). In this case, compared with the Type-C-I actuator, the decrease in the Type-C-II dynamic displacement is mainly because of the damping effect caused by the friction between the two adjacent actuation layers on each side, which are independent between each other in the current design (Supplementary file Text S1). The friction could be eliminated by combining the independent actuation layers into one multiple-layer actuation structure to reduce the damping effect and increase the dynamic displacement, which is an established method in literature that can be implemented in future work[32]. More importantly, it can be found that the resonant frequency of the proposed TS-DEAs can be easily tuned by changing the stiffness of the tensioning mechanism while not needing to change the properties of the elastomers by adjusting their recipe.

Based on the characterisation of the proposed TS-DEAs design, we calculated the output power and energy conversion efficiency[41,42] of these TS-DEAs across a range of frequencies. The calculations, along with the methodology, are detailed in Supplementary file Text S8.

Further, we conducted ageing tests on the Type-A and Type-C-I TS-DEAs to evaluate their performance changes over a period of fifteen days. The Type-A TS-DEA, utilising VHB as the elastomer material, is known for its high viscoelastic properties and showed a gradual and prolonged reduction in displacement in the first ten days before stabilising. In contrast, the Type-C-I TS-DEA, made from a silicone-based material, demonstrated a quicker reduction in displacement within the first five days and then stabilised, as detailed in Supplementary file Text S9.

## Locomotion tests of single TS-Robots

Crawling, climbing, steering, swimming and landing are five fundamental locomotion modes that single TS-Robots can perform, within and across solid and liquid domains.

We first tested the Type-A TS-Robot (Poisson's ratio $\nu = 0$) for crawling on horizontal surfaces and climbing on vertical walls with obstacles in the path such as, 2 mm high gaps (Fig. 3a and b, Supplementary Movie 3). The Type-A TS-Robot (1.7 mm high, 1.4 g) includes a Type-A TS-DEA and two EA-Pads (its fabrication and assembly process are presented in Supplementary file Text S6).

Figure 3c presents the gait and control strategy of the Type-A TS-Robot for crawling locomotion. Here, the TS-DEA was driven by a sinusoidal-wave voltage ($V_{D1}$, the solid-line signal), while the front and rear EA-Pads were driven by square-wave voltages ($V_{E1}, V_{E2}$), respectively. In a period of time, T, the robot moves forward in two steps. In the first step (0-T/2), the rear EA-Pad ($V_{E2} > 0$ kV) is actuated to adhere to the substrate while the front EA-Pad ($V_{E1} = 0$) is released. Simultaneously, the TS-DEA ($V_{D1} > 0$ kV) extends to push the front EA-Pad forward. In the second step (T/2-T), the front EA-Pad ($V_{E1} > 0$ kV) sticks to the substrate while the DEA contracts to move the rear pad forward ($V_{D1} = 0$; $V_{E2} = 0$). For crawling backwards, the phase of the sinusoidal-wave voltage ($V_{D1}$) is shifted by T/2 from the original signal (forward crawling—the solid-line signal, $V_{D1}$; backwards crawling—the dashed signal, $V_{D1}$)).

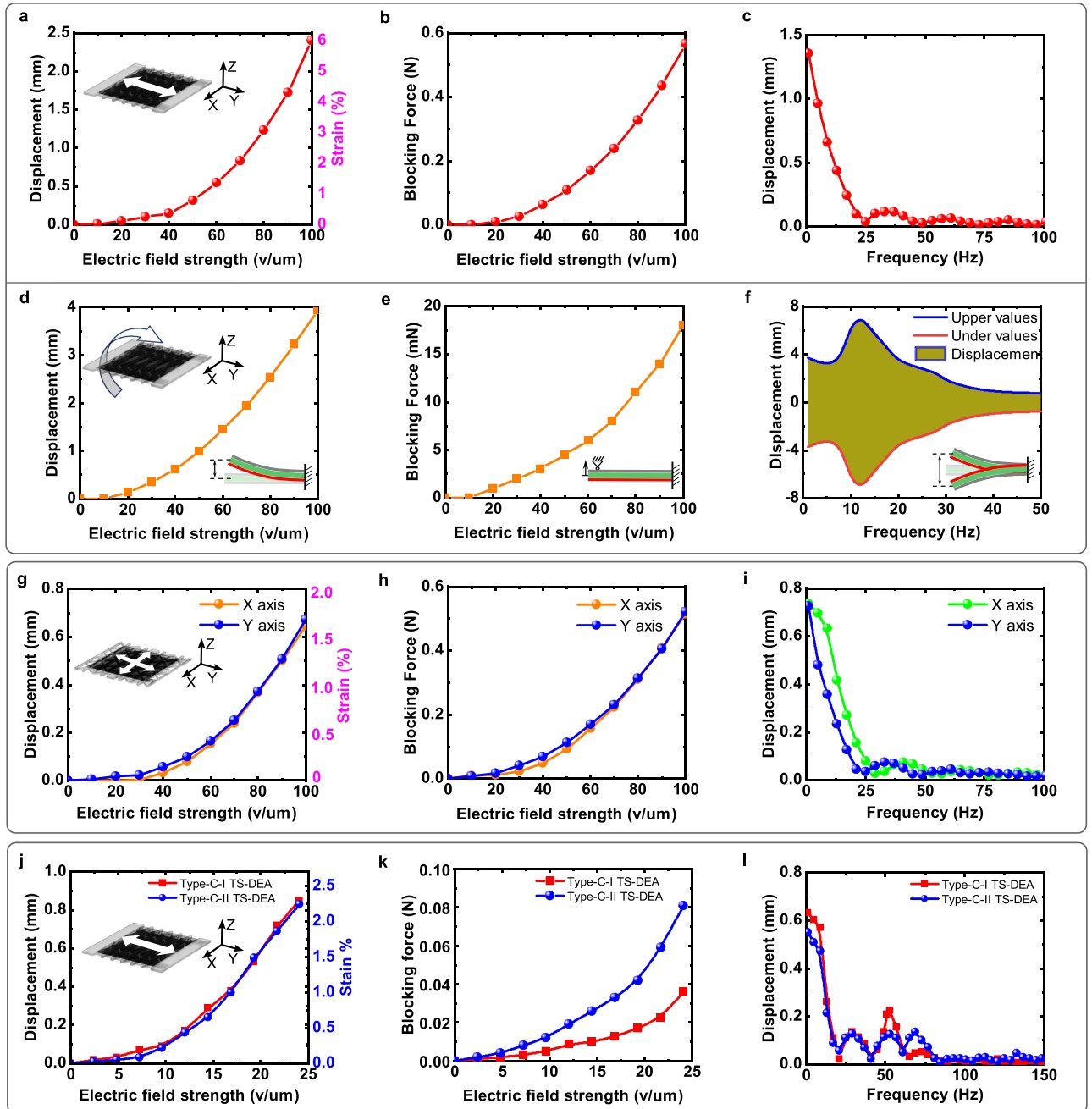

**Fig. 2 | Characterisation results of the three types of TS-DEAs. a** Static displacement of the Type-A TS-DEA vs. Electric field strength; **b** Blocking Force of the Type-A TS-DEA vs. Electric field strength when the actuator is constrained to its initial length; **c** Dynamic displacement of the Type-A TS-DEA vs. Actuation frequency (100 V/um); **d** Static bending displacement of the Type-A TS-DEA vs. Electric field strength; **e** Blocking Force of the Type-A TS-DEA bending vs. Electric field strength when the actuator is constrained to its initial shape; **f** Dynamic bending displacement of the Type-A TS-DEA vs. Actuation frequency (100 V/um); **g** Static displacement of the Type-B TS-DEA in X and Y axes vs. Electric field strength; **h** Blocking Force of the Type-B TS-DEA vs. Electric field strength when the actuator is constrained to its initial length; **i** Dynamic displacement of the Type-B TS-DEA vs. Actuation frequency (100 V/um); **j** Static displacement of the Type-C TS-DEAs vs. Electric field strength; **k** Blocking force of the Type-C TS-DEAs vs. Electric field strength when the actuator is constrained to its initial length; **l** Dynamic displacement of the Type-C TS-DEAs vs. Actuation frequency (21.7 V/um).

The crawling speed of the robot is mainly determined by the actuator's frequency-dependent displacement and the EA-Pad adhesion performance. The robot was tested on surfaces of different materials including, PET (polyethylene terephthalate), wood, paper and PVC (Polyvinyl chloride), to understand the effect of the substrate material on its speed when it was driven by different frequencies. The horizontal and vertical locomotion speeds of the robot on different surfaces are presented in Fig. 3d and e. It can be found that the robot can achieve its maximum speed of 2.3 mm/s (3.5% Body Length/s (BL/

s)) and 1.7 mm/s (2.8% BL/s) on horizontal and vertical PET surfaces, respectively, at a driving frequency of 4 Hz.

As an example of real-world applications of the proposed TS-Robots, we proved the concept of using a TS-Robot to inspect electrical generators for Rolls-Royce electrical aeroengine (Fig. 3f). In this demonstration, a variant of the Type-A TS-Robot (Curved-Type-A TS-Robot, denoted as C-Type-A TS-Robot, 0.8 mm thickness) was tested to evaluate the feasibility of in-situ inspections of the magnetic field degradation of an electrical generator for aircraft propulsion (i.e. an actual trend to

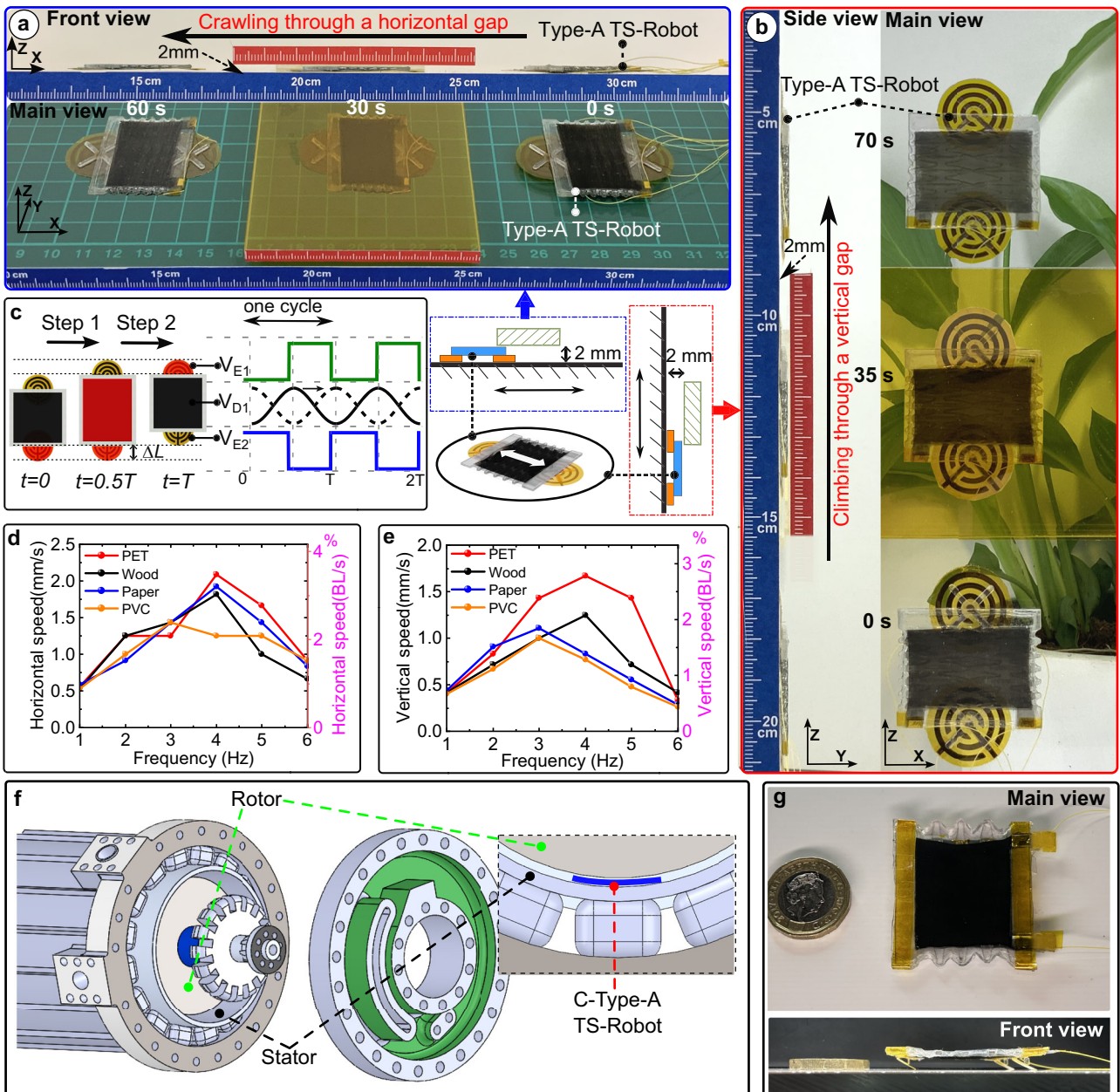

**Fig. 3 | Locomotion test of the TS-Robots (Poisson's ratio, zero). a** Test of the Type-A TS-robot crawling through a gap of 2 mm height on a horizontal transparent PET plate (yellow colour); **b** Test of the Type-A TS-robot climbing through a gap of 2 mm height on a vertical transparent PET plate (yellow colour); **c** Gait and control strategy of the Type-A TS-robot (the parts in red denote an activated actuator or EA-pads); **d** Locomotion speeds of the robot when crawling on different materials vs. actuating frequencies; **e** locomotion speeds of the robot when climbing on different materials with a range of frequencies. **f** Diagram of C-Type-A TS-Robot for inspection of the 2.5 MW generator; **g** Prototype of L-Type-C TS-Robot.

reduce CO2 emission globally), through the air gap between its rotor and stator. The motivation for the inspection is that the electrical generator for electrical-powered aero-engines (e.g., the 2.5 MW generator for the Rolls-Royce AE2100 hybrid power system) features a rotor with permanent magnets, and the properties of these magnets degrade over time, necessitating regular in-situ surveys to ensure the safety and efficiency of the generator with minimal financial and time cost. However, the 1.2 mm-high air gap between the rotor and stator renders it inaccessible for conventional tools, including rigid and soft robots. Here, we demonstrated the C-Type-A TS-Robot (Supplementary file Text S10) for locomotion in the narrow air gap of the Rolls-Royce AE2100 hybrid power system mock-up (Supplementary Movie 4). Through this demonstration, we show the TS-Robot could be used to address the challenges of accessing extremely narrow gaps in high-value industrial systems.

Silicone-based TS-DEAs provide outstanding speed performance due to their capability for high-frequency actuation. We demonstrate the enhanced mobility of the Type-C-II TS-Robot, which is equipped with three-directional friction feet, as detailed in Supplementary file Text S11. The robot achieved a maximum speed of 43.1 mm/s, corresponding to 1.16 times its body length per second and 13.1 times its body thickness per second, as shown in Supplementary Movie 5. The robot was actuated at its resonant frequency (67 Hz), which can be further increased by adding more actuation layers, thereby enhancing its speed, as clarified in Supplementary file Text S1.

Further, we tested a new version of one-DoF TS-Robots that operates at a low voltage of 220 V, named Low-Voltage-Type-C TS-Robot (L-Type-C TS-Robot, Fig. 3g), to showcase its potential for power autonomy. Actuated at its resonant frequency (86 Hz), the robot

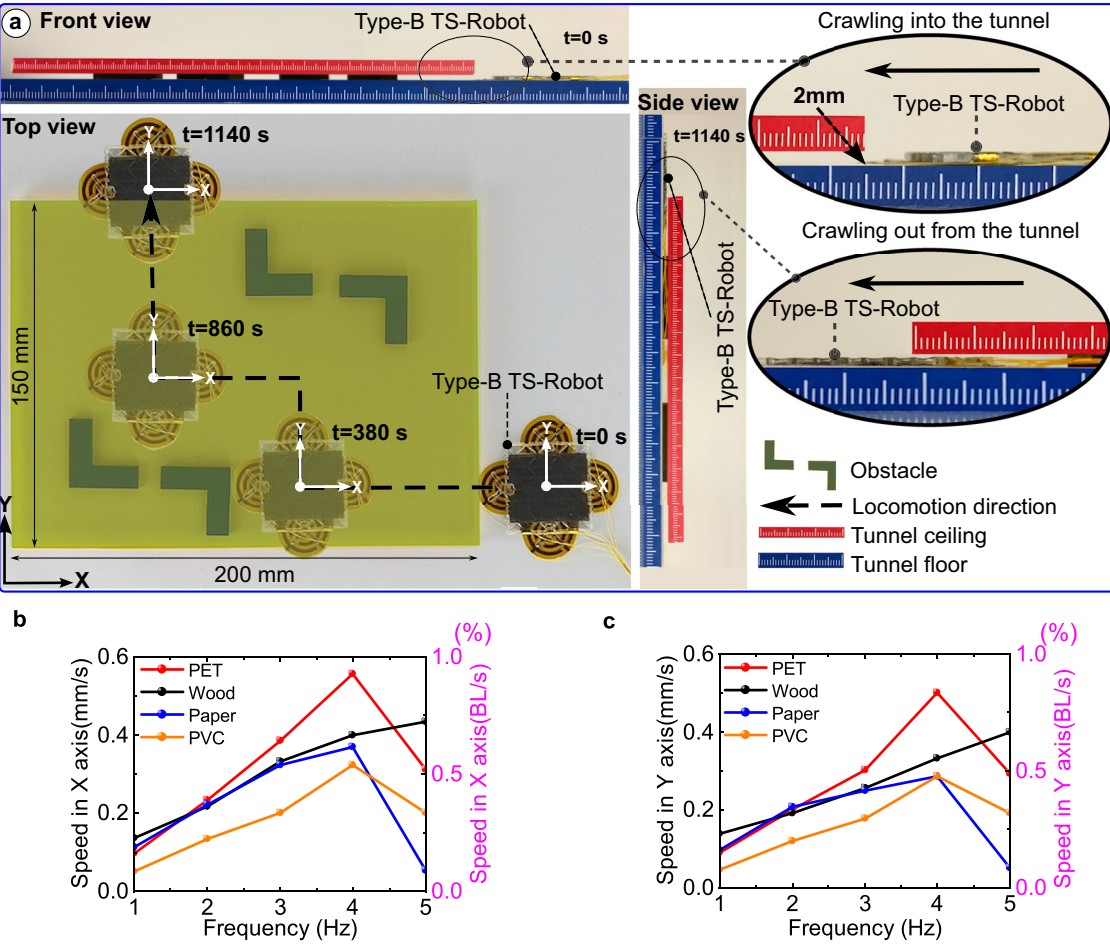

**Fig. 4 | Locomotion test of the Type-B TS-Robot. a** A test of the robot for crawling and steering through narrow spaces (2 mm high gap) containing multiple obstacles; **b** Speeds of the robot crawling along *X* axis on horizontal planes with different materials; **c** Speeds of the robot crawling along *Y* axis on horizontal planes with different materials.

reached a maximum crawling speed of 12.4 mm/s, equivalent to 0.33 times its body length per second and 3.75 times its body thickness per second (Supplementary Movie 6). We also demonstrate the load-carrying capacity of the L-Type-C TS-Robot, which can carry up to 3 g (2.5 times its self-weight), with results presented in Supplementary file Table S4.

When the narrow gaps contain insular obstructions/blockages, the Type-B TS-Robot (thickness, 1.7 mm; weight, 1.5 g), which is capable of steering in two directions, is used (Supplementary Movie 7). Figure 4a presents the test of the robot for crawling and steering through a 2 mm high tunnel (length: 200 mm, width: 150 mm) containing multiple obstacles. Starting at the lower right corner of the tunnel, the Type-B robot can change its crawling direction to avoid obstacles for moving to the upper left corner of the tunnel. It took the robot 1140 s to crawl a 350 mm long path to pass through the narrow tunnel. As Type-B TS-DEA can simultaneously extend in X and Y directions, the crawling direction of the robot was controlled by the synergy between the EA-Pads and DEA (Supplementary Fig. S23). Here, we tested the Type-B TS-Robot for performing crawling locomotion on the surfaces of different materials. It was found that the speed of the Type-B TS-Robot (1–5 Hz driving frequency) shares the same trend with the Type-A TS-Robot when crawling on surfaces of different materials (Fig. 4b and c). When the driving frequency is 4 Hz, the robot reaches a maximum speed of 0.57 mm/s and 0.51 mm/s in the X and Y axes on a horizontal PET plate, respectively.

When our crawling robot encounters liquid obstacles, the motion of the leech is imitated by using an undulating gait for swimming

through these barriers. We systematically explored the locomotion capability of the TS-Robots for swimming and transitioning between solid and liquid domains (e.g., oil and water). Different from the terrestrial locomotion, we use the bending motion of the TS-Robots, on which one of the EA-pads as a tail for generating thrust, to imitate the swimming motion of the leech. The Type-A TS-Robot (Poisson's ratio $v$=0) was used as an example with a small modification by moving the rear EA-Pad 5 mm backward to increase its swimming speed in liquid environments (Fig. 5a and Supplementary Movie 8). With the maximum voltage input of 5 kV and a driving frequency of 12 Hz, the robot can swim with an average speed of about 45.5 mm/s (0.7 BL/s) in liquid (Silicon oil, viscosity: 5, Supplementary Fig. S24), nearly 20 times faster than in terrestrial environments (2.3 mm/s on horizontal surfaces). One of the reasons is that a higher driving frequency was used for swimming (12 Hz) compared with crawling (4 Hz), as the maximum frequency for crawling was limited by the charging and discharging time of the EA-Pads (100–300 ms per cycle). By using a high-speed camera, we observed the undulating gait of the robot, when driven by 12 Hz swimming in the liquid environment (Silicon oil, viscosity: 5; Fig. 5b, Supplementary Movie 8). Further, we demonstrated that the TS-Robot can be operated in aquatic environments (0.8 BL/s), two wiring methods were proposed and tested: the two-pole and three-pole wiring methods. (Supplementary file Text S12 and Supplementary Movie 8).

To illustrate the cross-domain locomotion capability of the Type-A TS-Robot, we demonstrated that the robot uses two locomotion modes (crawling and swimming) in a hybrid solid-liquid environment (Fig. 5c, Supplementary Movie 9). In the experiment, the robot started

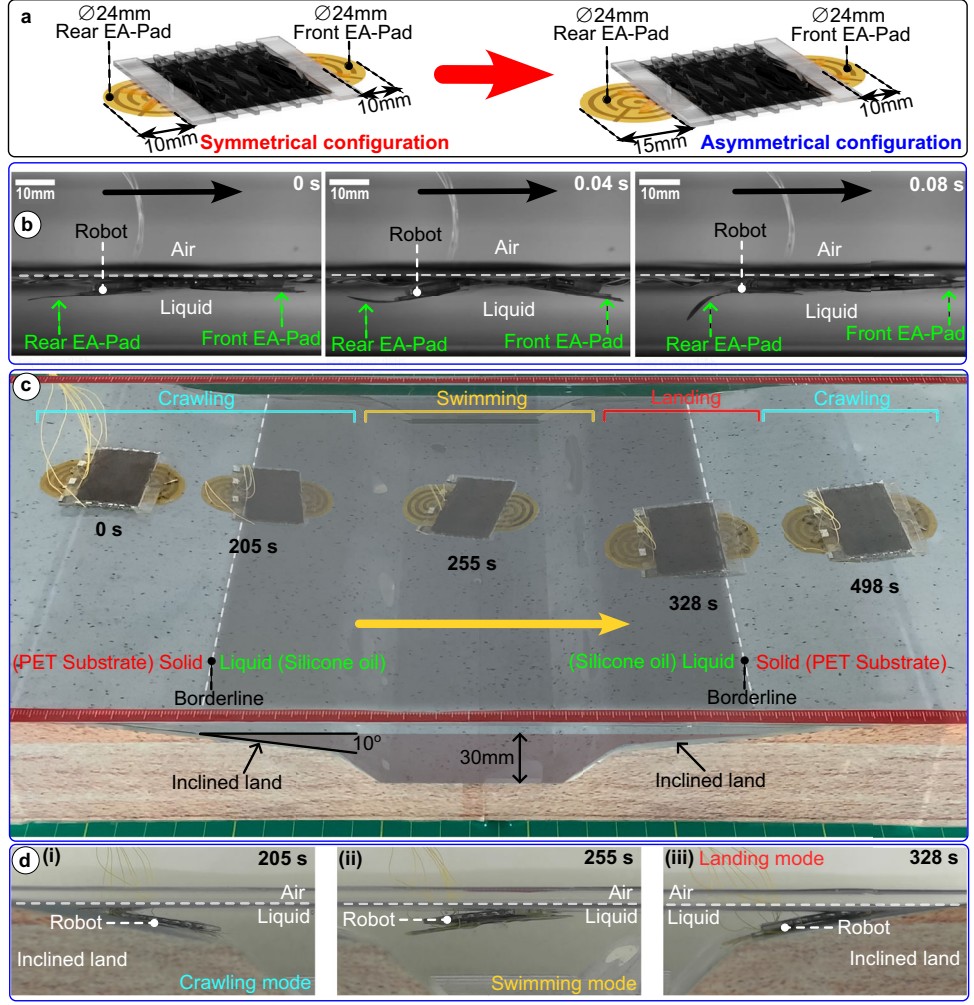

**Fig. 5 | Swimming and Cross-domain locomotion test (solid-liquid domains) of the Type-A TS-Robot. a** The modification of the Type-A TS-Robot for efficient swimming; **b** High-speed camera views of one cycle of the robot's undulating gait during swimming; **c** A demonstration of locomotion of the Type-A TS-Robot in a hybrid solid-liquid environment; **d** Side view of the three stages of the cross-domain (solid-liquid-solid) locomotion demonstration.

to crawl from an obtuse-angle corner (170°) onto an inclined dry surface (10°) until it was submerged in the liquid (Fig. 5di). By discharging the EA-Pads, the robot floated from the bottom of the liquid to the surface, while switching from crawling to swimming locomotion (Fig. 5dii). From 328 s, it was found that the robot propelled itself forward by using the rear EA-Pad as its swimming tail and transitioned from being submerged in liquid to semi-submerged (Fig. 5diii). Once both EA-Pads were electro-adhesively attached to the substrate, the robot switched to the crawling mode and moved onto the dry surface.

In these sets of experiments, we presented the crawling, climbing, steering, swimming, and cross-domain (hybrid liquid-solid environment) locomotion of the TS-Robots to highlight the abilities of single thin-soft robots.

**Locomotion and manipulation test of multiple joined TS-Robots**

When the required tasks present unfeasible challenges for single TS-Robots such as, transitions between surfaces and complex manipulations, our proposed TS-Robots can be joined together as "an assembly" to execute the mission collaboratively. Firstly, to transition between surfaces, we combined two Type-A robots connected with a passive hinge joint, as a twin system, with a pair of "pre-bent whiskers" equipped at the front robot (Fig. 6a). The twin system cannot only access narrow spaces but is capable of transitioning between horizontal and vertical surfaces. With the collaboration of two robots, the

twin Type-A TS-Robot can perform three gaits (Fig. 6b and Fig. S25), Gait I: both robots crawl/climb on horizontal/vertical surfaces, Gait II: the rear robot pushes the front one onto another surface (e.g., wall) with the guidance of a pair of "pre-bent whiskers" and Gait III: the front robot pulls the rear one onto another surface (e.g., wall).

By using this set of gaits, we demonstrated that our twin thin-soft robotic system can advance through a more challenging built environment by crawling on a flat horizontal surface (Gait I), then transitioning onto a vertical surface by collaborations between them (Gait II), and finally climbs on the vertical surface, (Gait III) (Fig. 6c and Supplementary Movie 10).

Further, in order to enhance the manipulation and steering capability of the twin thin-soft robot, a Type-B TS-Robot (Poisson's ratio $v = -1$, two-directional movement) was used to replace one of the Type-A robots; Type-B and Type-A TS-Robots are serially connected by using an active hinge joint (Fig. 6d). We call the new system a serial kinematic TS-Robot (SK-TS-Robot, Supplementary file Text S13). By integrating an active hinge joint (a biasing bending DEA, Supplementary file Text S14) into the SK-TS-Robot, the system can switch between a flat state when the active hinge joint is actuated and a "-Shape state when it is not actuated. Thus, the design of the SK-TS-Robot can perform four gaits for crossing through narrow gaps, steering, transitioning between surfaces and operating as a manipulator (Fig. 6e and Supplementary Fig. S26):

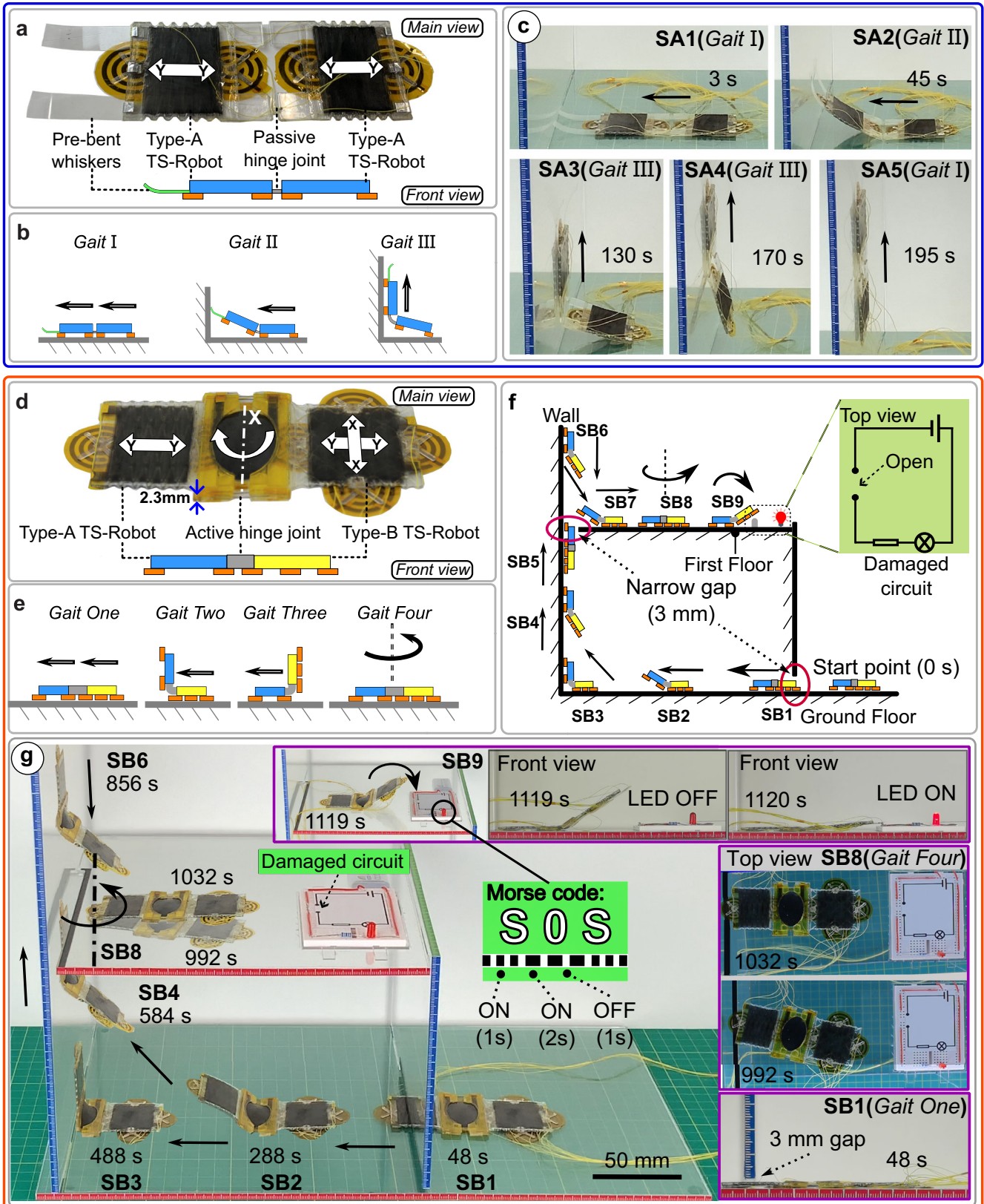

**Fig. 6 | Locomotion and manipulation test of the multiple joined TS-Robots.**
**a** The structure of the Twin Type-A TS-Robot; **b** Three gaits of the Twin Type-A TS-Robot; **c** Transition between horizontal and vertical surfaces by collaboration between two Type-A TS-Robots; **d** The structure of Serial Kinematic TS-Robot (SK-TS-Robot); **e** Four gaits of the SK-TS-Robot; **f** Diagram of the robot locomotion path in the two-floor test rig; **g** Selected steps in the demonstration of the robot locomotion in the two-floor test rig.

i) Gait One: Flat Linear Crawling, where the SK-TS-Robot moves along its length direction ($Y$ axis); both the Type-A and Type-B TS-Robots were actuated to crawl in their Y axes.

ii) Gait Two: Front Flip-Up, where the SK-TS-Robot is an "L" Shape state, and the Type-B TS-Robot moves along its $Y$ axis when performing as a "carrier" of the Type-A TS-Robot and the active hinge joint.

iii) Gait Three: Back Flip-Up, where the SK-TS-Robot is an "L" Shape state but with the Type-B TS-Robot lifted while the Type-A TS-Robot is used as a "carrier" of the entire system.

iv) Gait Four: Steering where the SK-TS-Robot is in the "flat" state and the Type-B TS-Robot is driven to crawl in its $X$ axis, resulting in a turning motion of the entire robot, caused by the friction between the robot and the supporting surface.

To demonstrate the locomotion and manipulation abilities of the SK-TS-Robot, we built a two-floor test rig (Fig. 6f) containing two narrow gaps (3 mm high): one horizontal on the ground floor; one vertical on the first floor. The SK-TS-Robot started at the entrance on the ground floor and, through nine steps (SB1−SB9) on which all four gaits were used, it reached the first floor and re-connect a "damaged" electric circuit (see the indication in Fig. 6g) with its front EA-Pad while sending programmable light singles (e.g., SOS) via an LED of the circuit, by selectively connecting the circuit in a periodic manner.

Figure 6g presents several selected steps of the process in the trial (Supplementary Movie 11). In Step One (SB1), the SK-TS-Robot started at the entrance of the test rig in a flat configuration, advancing through a 3 mm high narrow gap by Gait One (Flat Linear Crawling). After passing through the horizontal narrow gap, the active hinge joint was actuated for lifting the Type-A TS-Robot, as the Type-B robot kept crawling forward as a "carrier" by Gait Two (Front Flip-Up, SB2). When the robot reached the vertical wall, the EA-pads of the Type-A TS-Robot were activated to adhere to the wall, while those of the Type-B system were deactivated to allow the entire system to detach from the ground floor afterwards (SB3). Then, the SK-TS-Robot climbed onto the wall using Gait Three (Back Flip-Up) to transition from the horizontal surface to the vertical wall, which employs the Type-A TS-Robot as a "carrier" of the entire system (SB4). Once the robot climbed higher than its own length on the wall, it returned to the flat configuration and then advanced through the gap at the conjunction between the wall and the first floor by using Gait one (SB5). After passing the vertical narrow gap, the active hinge joint was used to lift the Type-B TS-Robot when the Type-A TS-Robot moved downward until the Type-B robot was attached to the first floor (SB6). At the conjunction, the Type-B robot performed as a "carrier", while the Type-A one was disengaged from the wall, followed by the entire system returning to the straight state (SB7-Gait Two). Then, the robot adjusted its crawling direction by Gait Four (Steering) and advanced to the target zone, a "damaged" electric circuit by Gait Two (SB8). Once the robot reached this target zone, the active hinge joint was actuated in a controlled manner to connect the "damaged" circuit by using one of the EA-Pads in the Type-B TS-Robot, which turned the LED on. We further demonstrated this manipulation capability of the robot by sending programmable light signals ("SOS" Morse code) via the actuation of the active hinge joint (SB9). During the whole process, the maximum speed of the SK-TS-Robot was 1.18 mm/s when climbing downwards on the wall (SB6- Gait Three), while the minimum speed was 0.34 mm/s when the robot climbed upwards on the wall (Supplementary Fig. S27).

### Locomotion tests of TS-Robots collaborating with other types of robots

These TS-Robots can also collaborate with other types of robots as carriers and gain additional locomotion modes from the carried system for exploring more challenging environments.

As the TS-DEAs can generate large propelling forces compared with their own weights (e.g., Type-A DEA: maximum 0.57 N, 48 times its

weight), the TS-Robots can be used to carry other robots such as, small drones (Fig. 7a), enabling the hybrid systems to crawl and fly for operating in solid and aerial domains and transitioning between them. In this test, we used a hybrid system (Flying-TS-Robot), composed of a Type-A TS-Robot and a small off-the-shelf drone (Fig. 7b), for delivering objects. For connecting the robot and the drone, an EA-Pad (E2) is located on the top of the Type-A TS-Robot, and then the two are connected by switching on the charge of the EA-pad. The other EA-Pad (E1) is integrated into the front of the robot via a thin beam (0.5 mm thick PETG) for attaching (via electro-adhesion) and carrying light objects (e.g. letters) for delivery tasks (Fig. 7b).

We tested the Flying-TS-Robot for delivering a "letter" into an enclosed space containing a narrow gap that is separated from the start point by a deep trench to demonstrate the cross-domain locomotion capability of the system (Fig. 7c, Supplementary Movie 12). In Step SC1, the TS-Robot (1.4 g), carrying the drone (36 g), crawled 200 mm through a 50 mm high tunnel in 223 s (Fig. 7d). Note that the height of this tunnel proved to be too small to enable the drone to pass by flying (as demonstrated in Supplementary Movie 12); as such, the Flying-TS-Robot crawled through it by using the Type-A TS-Robot attached to the drone. Next, this hybrid system flew together to cross a deep trench quickly (Step SC2- in a duration of 2 s). Note that without the drone, the TS-Robot cannot cross over to access the delivery point because of the trench. After landing, the crawling robot and the drone were disengaged by discharging the EA-Pad (E2) thus allowing the drone to release the TS-Robot (Step SC3). The Type-A TS-Robot then crawled through a 3 mm high gap while carrying the "Letter" (Step SC4, Fig. 7e) electro-adhesively attached to it, as previously mentioned. Finally, the EA-pad (E1) was discharged to release the "Letter" in Step SC5, and the TS-Robot crawled backwards (Fig. 7f). In this test, the crawling TS-Robot used the gait presented in Fig. 3c. To enable a conscious gait selection for the hybrid robotic system, we tested the crawling speeds of the Type-A TS-Robot with various payloads at different frequencies (Fig. 7g, and Supplementary Movie 13). It can be found that the robot speed decreases as its payload increases (from 0 to 40 g) due to the increased friction between the EA-Pad and the substrate, which matches the characterisation results of the Type-A TS-DEA. Based on the test result, we used an actuation frequency of 3 Hz to achieve an optimal speed (1 mm/s) for the Type-A TS-Robot carrying the drone.

Through these experiments, we show that the TS-Robot can carry other robots, such as a commercial drone, for terrestrial locomotion in confined space, and the hybrid system can perform cross-domain locomotion in both terrestrial and aerial environments. Additionally, we demonstrated collaborations between a Flying-TS-Robot and SK-TS-Robot for delivering "parcels" in a domestic scenario (Supplementary Movie 14).

## Discussion

The design and fabrication of thin DEA-driven robots for multimodal locomotion are challenging, especially when accessing near-millimetre and submillimetre-high narrow spaces, which are the targeted application scenarios in this paper. While multi-modal locomotion in free/ large spaces has been achieved by many existing robots, when it comes to designing a thin soft robot for accessing narrow spaces in built environments and performing multimodal locomotion, significant challenges need to be addressed:

1. Challenge 1 (Ch1): locomotion in multiple domains (e.g. solid/ liquid media) in narrow spaces often requires a robot with a thin thickness to perform multiple gaits[43], such as linear gait for crawling on solid surfaces or through narrow gaps and undulating gait for swimming in liquid. In contrast, a robot with a single gait may perform locomotion in one domain but may find it challenging in another[20], or it may perform multiple gaits but need to be actuated by an external magnetic field, which could be challenging to be used in built environments[43].

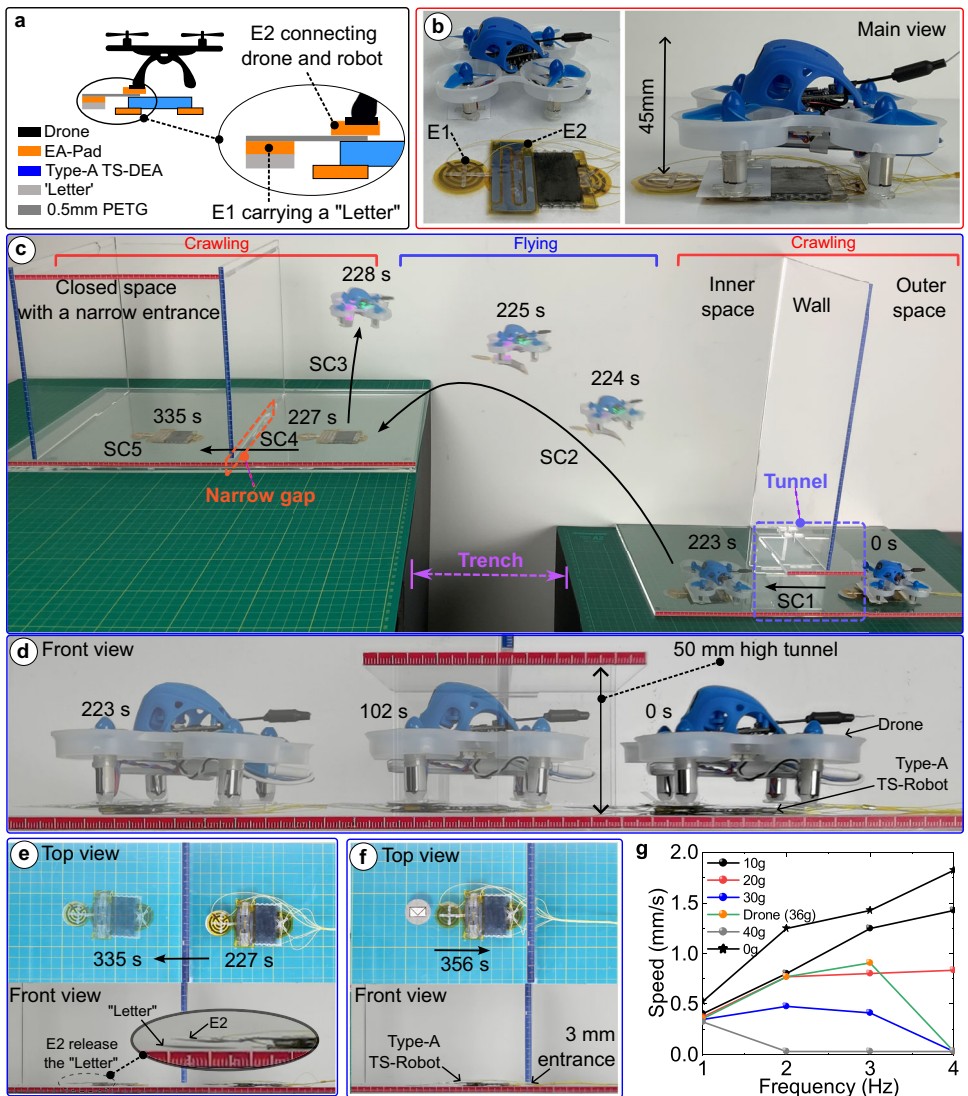

**Fig. 7 | Cross-domain locomotion (Terrestrial-Aerial) in narrow spaces by collaborations between Type-A TS-Robot and a drone. a** Schematics of a hybrid system containing a drone and a Type-A TS-Robot; **b** Prototype of the hybrid system, a Flying-TS-Robot; **c** Terrestrial-Aerial locomotion performed by the Flying-TS-Robot for delivering a "Letter" in a confined space; **d** The experiment of the TS-Robot carrying the drone and crawling through a narrow tunnel; **e** The TS-Robot crawls through a narrow gap carrying a "Letter"; **f** The TS-Robot moves backwards after the Letter released; **g** Speed of the TS-Robot when carrying payloads in different frequency.

2. Challenge 2 (Ch2): Multimodal locomotion in narrow spaces may need several robots to work together to transition between different terrains[44], such as transitioning between horizontal and vertical surfaces and crossing large gaps. This requires the robot to have a high payload/self-weight ratio and stiffness to carry another robot to assist each other in crossing the barriers.

Thanks to the proposed structure of the TS-DEA, these robots can be built with near-millimetre and submillimetre thickness, and achieve two types of gaits, i.e., a linear in-plane elongating/shortening and an out-of-plane bending gait (Challenge One). Based on this design, the TS-Robots with thin profiles can independently cross various obstacles (e.g., narrow gaps, walls, liquid pools) while operating in solid and liquid domains or transitioning between them. We demonstrated that a single Type-A TS-Robot performs multimodal locomotion, including crawling, climbing, steering, swimming, and landing, for exploring environments containing multiple types of obstacles and even multiple domains. Further, the TS-DEA can be designed with more DOFs and anisotropy mobility in different axes by tuning the Poisson's ratio of the tensioning mechanism (e.g., Type-B TS-DEA).

More importantly, the TS-Robots can also be joined among themselves to work together for transitioning between surfaces (e.g., wall and floor) and operate as manipulators (e.g., repairing circuits and sending programable signals), because of their high payload/self-weight ratio (maximum 29:1) and stiffness (311.5 N/m and 4.5 N/m for linear and bending movements) (Challenge Two). In addition, we demonstrated that the TS-Robots can collaborate with other types of robots as "carriers", for example, by carrying a miniaturised drone with a TS-Robot, enabling both the "carrier" and "passenger" with extra locomotion mode (e.g., Drone–flying over deep trenches; TS-Robot: pass through difficult-to-access spaces). We envisage that such robots are promising solutions for several real-world applications, including inspection of confined spaces within industrial plants and civil engineering buildings, performing security surveys of confined spaces, as well as accessing and observing hazardous environments (Fig. 8).

Further, the proposed design enables a new method for tuning dielectric elastomer actuators' resonant frequency by mechanically adjusting the tensioning mechanism's stiffness. Compared with the state-of-the-art method (a chemical approach[32],), this mechanical method could be less time-consuming for turning the DEA properties

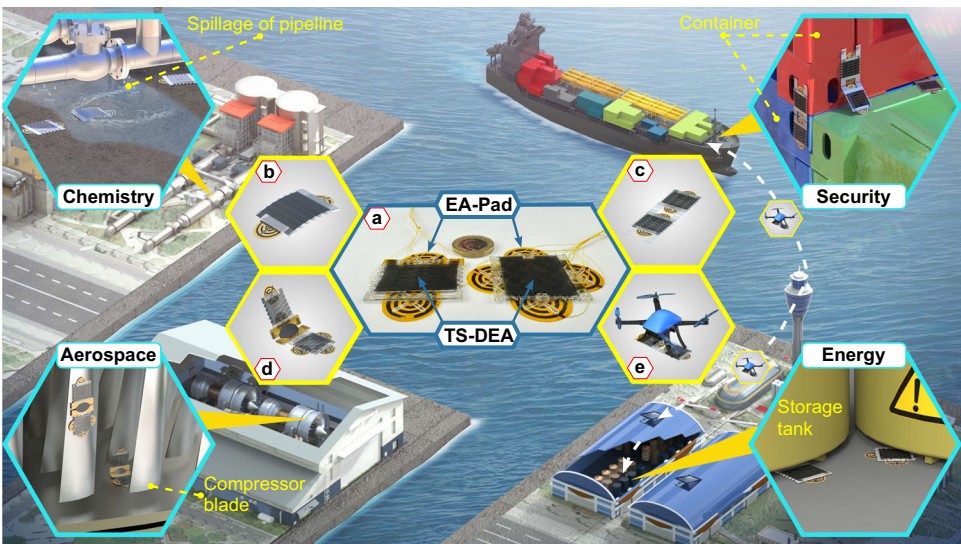

**Fig. 8 | Concepts of the Thin-Soft Robots (TS-Robot) for potential applications.** **a** Type-A TS-Robot (left) and Type-B TS-Robot (right), targeting for navigation in narrow gaps on horizontal or vertical surfaces ("Energy" scenario); **b** Asymmetric configuration of Type-A TS-Robot, targeting for the hybrid solid-liquid environment ("Chemistry" scenario); **c** Twin Type-A TS-Robot, targeting for transitions between horizontal and vertical surfaces ("Security" scenario); **d** Serial Kinematics TS-Robot, targeting for transitions between horizontal and vertical surfaces ("Aerospace" scenario); **e** One example of TS-Robots collaborating with other types of robots−Flying-TS-Robot, targeting for Transitioning between surfaces by using a carrier ("Security" and "Energy" scenario).

to achieve the targeted resonant frequency. This is a fundamentally new method for improving DEA performance, which can be applied to a wide range of DEA designs. The Type-C TS-Robot has demonstrated outstanding performance in speed, reaching 43.1 mm/s (Body Length 37 mm, Body Thickness 3.3 mm), which is 1.16 times Body Length/s and 13.06 times Body Thickness/s.

In the field of soft robotics, thin robots can also be achieved by different actuation methods, such as electrostatic[13] and piezoelectric[45], which have been used to build insect-scale robots[45] for fast locomotion on flat surfaces. However, dielectric elastomer actuators (DEAs) offer significant advantages such as high flexibility, large strain, and high power density, which enables multi-modal locomotion by single or multiple robots. Compared with the state-of-the-art small-scale robots (Table 1), the Thin Soft Robots reported in this work are more suitable for exploring complex-built environments, especially when needing to access near-millimetre and submillimetre-high narrow spaces, such as the gaps between components in large machines or industrial plants. In addition, to the authors' best knowledge, this is the first small-scale robot capable of collaborating with drones, enabling both with an additional crawling locomotion mode.

The limitations of the current robots can be summarised as follows: the dielectric elastomer actuation layer is made of a single layer of DEA, limiting the maximum amplitude of the force output, which can be enhanced by using multiple-ultrathin-layer DEA made from advanced fabrication approaches[46,47]. Further, the TS-robots were proposed to explore narrow spaces in high-value industrial equipment/environments, which can be used in a wide range of real-world applications, including, but not limited to, the inspection and maintenance of industrial equipment such as aero-engines, aircraft, and nuclear facilities. Due to the high-value nature of these equipment, robots are often required to be tethered for robust communication and safety in these confined spaces to avoid communication failures and retract the robots if they fail. Hence, the current robot is powered and controlled through thin wires. However, it may limit the reach range of the robots due to the constraints of the wire length, which can be addressed by using a battery-powered untethered design containing ultra-thin-layer DEAs.

## Methods
### Materials and fabrication of the TS-DEAs
Two types of elastomer materials were used to construct the TS-DEAs. The VHB-based elastomer (VHB 4910) was used for Type-A and Type-B TS-DEAs, while silicone-based elastomers were utilised for Type-C TS-DEAs. Specifically, a mixture of Ecoflex 00-30 and Sylgard 184 in a 1:1 ratio, with Sylgard 184 prepared at a cross-linker ratio of 40:1, was used for Type-C-I and Type-C-II TS-DEAs. Pure Ecoflex 0030 was employed for the L-Type-C TS-DEA. Both silicone-based materials were fabricated using a spin coater (WS-650Mz-23NPPB, Laurell). Each actuation layer comprised two electrodes (Thermo Scientific, multi-walled carbon nanotube: 8 nm outer diameter, 2-5 nm inner diameter, 0.5−2 micron long). An ultra-thin VHB elastomer (3 M, VHB 9460) served as the adhesive layer for bonding actuation layers to the tensioning mechanism in VHB-based TS-DEAs, while Bond flex was used for the silicone-based TS-DEAs. The tensioning mechanisms were made from PETG sheets, typically 1 mm thick, while the C-Type-A TS-DEA utilised a thinner 0.25 mm PETG sheet, aiming for a thinner DEA/Robot, all manufactured by a laser cutter (Jindiao Technology Co., Ltd, JD3050). For detailed design and step-by-step fabrication procedures of TS-DEAs, refer to Supplementary file Text S1, Text S6 and Text S13.

The terminal of the DEA is constructed using Aluminium foil tape (3 M, 1436, 0.38 mm thickness). Via the terminals, the DEA is connected to the control and power system via a thin lead wire (Daburn, Ultra-Flexible Microminiature FEP 150 V Lead Wire, 2420/42 YL−100).

### Materials and fabrication of the EA-pads
The EA-Pads were fabricated by printing the electrode onto a polyimide film (diameter: 25 mm; thickness: 0.025 mm) as the substrate (Supplementary Fig. S8). The silver electrodes were printed using a Fujifilm Dimatix Materials Printer DMP2850 Series and a 2.4 pL Samba cartridge loaded with the XTPL IJ-36 silver ink. The droplet diameter is 40 μm, and a drop spacing of 20 μm was used to ensure the deposition of continuous layers and pattern uniformity. Six layers were printed on a polyimide substrate, and the substrate was heated to 80 °C by an in-situ resistance heater. Then, it is post-processed at 150 °C for an hour. The thickness of each layer is 500 nm, and six layers were printed on the polyimide substrate, making the total thickness of the EA-pad

**Table 1 | Comparison of the locomotion modes and designs of soft robots**

| Soft robots | Actuation | Height (mm) | Locomotion types | Transition between surfaces | Transition between domains | Feasible for built environments |
|---|---|---|---|---|---|---|
| This work | Dielectric elastomer | 1.7 (Type-A and B) 0.8 (Type-C) | Crawling, Climbing, Steering, Swimming, Landing | Yes | Yes | Yes |
| Guo, 2020[19] | | 2.1 | Crawling | No | No | Yes |
| Ji, 2019[20] | | ~4.2 | Crawling, Steering | No | No | Yes |
| Tang, 2022[27] | | 9.8 | Crawling | No | No | Yes |
| Gu, 2018[26] | | 10* | Crawling, Climbing, Steering | No | No | Yes |
| Zhu, 2023[48] | | ~12 | Crawling, Steering | No | No | Yes |
| Duduta, 2020[46] | | ~15.9 | Crawling, Hopping, Jumping, Rolling | No | No | Yes |
| Hu, 2018[43] | Magnetics | 0.65* | Crawling, Walking, Swimming, Jumping, Rolling, Landing | Yes | Yes | No |
| Wu, 2022[49] | | 1* | Crawling, Climbing, Rolling, Walking, Swimming | Yes | Yes | No |
| Ze, 2022[9] | | 3.9 | Crawling, Steering | No | No | No |
| Mao, 2022[23] | | 6 | Walking, Crawling, Jumping, Swimming | Yes | Yes | No |
| Wu, 2023[21] | Thermal | 1* | Crawling | No | No | Yes |
| Pang, 2022[30] | | 6 | Crawling, Climbing | Yes | No | Yes |
| Wu, 2019[14] | Piezoelectric | 3 | Jumping, Crawling | No | No | Yes |
| Zhang, 2022[17] | Pneumatic | 80 | Crawling, Climbing | Yes | Yes | Yes |
| Wang, 2021[13] | Electrostatic | 2 | Crawling | No | No | Yes |
| Ahn, 2019[50] | Light | 31 | Crawling, Jumping, Squeezing | No | No | Yes |
| Zhu, 2021[51] | Chemical | 1 | Swimming | No | No | Yes |

Note: *The value refers to the minimal height of the confined space that the robot has been tested, as reported in the paper. Further, A mapping of the speeds of the state-of-the-art DEA-driven soft robots is shown in the Supplementary file Supplementary Fig. S29.

0.028 mm. The EA-Pad is supported by an X-shaped frame attached adhesively to the TS-DEA.

**Actuation and control system of the TS-Robots**

We used two types of control signals to actuate the VHB-based TS-Robots: a sinusoidal wave for the TS-DEAs, and a square wave for the active joint powered by dielectric elastomers and the EA-Pads, which have been illustrated in Fig. 3c, Supplementary Fig. S23, Supplementary Fig. S25 and Supplementary Fig. S26. A control system was developed to generate these signals for controlling the robots, as depicted in Supplementary Fig. S28. The sinusoidal wave was generated by a programmable high voltage converter (HVM Technology, Inc. HVA0560), which was directly controlled by an Arduino controller board (mega 2560). The square wave signal was derived from a DC-DC high voltage converter (XP power, Q101-5), which was controlled by an H-Bridge (L298N) driver board. To generate these signals, we utilised two programmable high-voltage converters for powering the TS-DEAs along with seven DC-DC high-voltage converters for the EA-Pads and the active joint. All these converters were synchronised through the Arduino controller board. For the silicone-based TS-Robots, the actuation signals were generated using a signal generator (Wave station 2022, Teledyne LeCroy) and then fed into high-voltage amplifiers (Model 610E-K-CE, Trek Inc.) to amplify the signals by a factor of 1000.

## Data availability

All data generated or analysed during this study are included in the manuscript and its Supplementary material. Source data are provided in this paper.

## Code availability

All code generated or analysed during this study are available through https://1drv.ms/f/c/29c50183fd5b1c7c/Eh7NkFRSBRlGtAwt82J0yswBeFwOEoOU2uGeH4ioFQtRVA?e=S0xPG1.

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

## Acknowledgements

We would like to acknowledge the support of Dr Feiran Wang of the University of Nottingham for printing the circuit of the EA-Pads. We also acknowledge Dr Ming Li from the University of Nottingham, for his assistance in characterising the electrical properties of the TS-DEAs. We thank Dr Ke Li from the University of Nottingham for the assistance in the design of the high-voltage and frequency power supply system. Further, we thank Mr. Samuel Wild and Mr. Nicholas Breeze from the Rolls-Royce University Technology Centre in Manufacturing and On-Wing Technology at the University of Nottingham for reviewing the manuscript. This work was also supported by the China Scholarship Council Research Excellence Scholarship with the University of Nottingham (Dr Xi Wang, Mr Siqian Li), EPSRC (EP/W001128/1) and Innovate UK (51689).

## Author contributions

In this paper, X.D., X.W., and D.A. conceived the idea. X.W. and X.D. designed Type-A, Type-B and Type-C TS-DEAs and TS-Robots and conducted their experiments. S.L. and X.D. designed the Flying-TS-Robot and conducted experiments on this robot as well as the swimming tests of the Type-A TS-Robot. X.W. and J.L. developed the electronics of the TS-Robots. X.W., J.C., and X.D. developed the theoretical models of the TS-Robots. X.W. and S.L. fabricated all the robots. X.D., X.W., and D.A. wrote the manuscripts.

## Competing interests

The authors declare no competing interest.
