## [Peer Review File · Nature Communications]

REVIEWER COMMENTS

Reviewer #1 (Remarks to the Author):

The authors present an interesting platform for robotics research: the tunable Poisson ratio for the dielectric elastomer actuators is a powerful tool to produce robotic function from design parameters. The results are compelling, as the robots can locomote between varying environments that have been challenging for similar machines in the past.

As it stands the work has three limitations which may prevent acceptance in this journal:

1. the use of VHB elastomer as the working material is disappointing: while the material is commercially available, it has major limitations particularly when it comes to fast actuation, due to its appreciable viscoelasticity. Considering the robotic applications, I urge the authors to consider other materials, including combinations of commercially available Ecoflex 00-30 and Sylgard 184, proven to actuate above 200 Hz. (Citation: Zhao, H., Hussain, A.M., Duduta, M., Vogt, D.M., Wood, R.J. and Clarke, D.R., 2018. Compact dielectric elastomer linear actuators. *Advanced functional materials*, 28(42), p.1804328.). However I understand that changing the elastomer might be reserved for future studies, beyond the scope of the current work.

2. the characterization of the actuators is incomplete: dielectric elastomer actuators are extremely interesting because of the direct conversion of electrical energy into mechanical work. Multiple studies have looked at maximizing the conversion efficiency, one good work can be found here: Bigué, J.P.L. and Plante, J.S., 2011. Experimental study of dielectric elastomer actuator energy conversion efficiency. *IEEE/ASME Transactions on Mechatronics*, 18(1), pp.169-177. The use of the controllable Poisson ratio opens up new possibilities in enhancing the energy conversion. However the energetics of the system are not studied almost at all by the authors. To strengthen this work, and show its clear advantages over state of the art, I encourage the authors to look at how much mechanical work is produced from the input of electrical energy, across ranges of frequency and operation modes. As it stands the robot seems to only be usable in a tethered fashion, so an understanding of energy is needed if it were ever to become autonomous.

3. Operation in the real world needs reliable and resilient robots. Unfortunately, no data is shown on how the performance of these robots changes over time or as a function of actuation cycles. Either would be extremely interesting, and enrich the field of soft robotics.

Reviewer #2 (Remarks to the Author):

In this paper, the authors describe a thin-film robot driven by dielectric elastomer actuators. The robot demonstrates multimodal locomotion such as crawling, climbing, and swimming. Overall,

the experimental demonstration is thorough, and the writing and figures are of high quality. However, in my opinion, this paper does not represent significant contributions to the microrobotic and soft robotic communities and I do not feel it is worthy of publication in Nature Communications. My main concerns and suggestions are described below.

Main concerns:

1. I do not feel the paper has addressed important and fundamental questions in the research fields. First, the performance of the reported robot is not outstanding compared to other DEA-driven robots. The crawling speed is particularly slow compared to the state-of-the-art results [1]. I cannot identify novelty in the DEA fabrication method, and the DEA performance (blocked force, maximum energy/force/displacement output) is not competitive. I do not see how this work can benefit future studies in improving DEA performance. Second, the DEA field is moving towards power autonomy [1], yet this robot is always tethered to an amplifier that commands 2kV- 5kV. Given the high operation voltage and small size, I think that there is a long way for this robot to achieve power autonomy. Third, the reviewer has experience working with aquatic microrobots. One of the biggest challenges is that water is conductive and it has non-negligible surface tension. I believe the authors use silicone oil as the liquid to bypass shorting problems. I do not think this is a good swimming demonstration considering the work has not addressed challenges in preventing shorting and overcoming surface tension. These three big challenges – DEA performance, power autonomy, and high voltage insulation – have not been sufficiently addressed.

2. One novelty of this work focuses on climbing via electrostatic adhesion, but the design is very similar to an existing work [2]. I also do not see significantly improved performance or new physical principles. I think enabling transitioning across a corner represents a new function, but it is incremental as it is not difficult to achieve in the prior robot [2].

3. The Discussion section lacks a detailed description of the work's implication and impact. Why are the chosen demonstrations difficult for existing robots and why are they important? What major bottlenecks have been solved in this work and what is the major novelty? I think the presented robot is still very far from being deployed in realistic applications, and I do not see how these demonstrations are challenging for existing robots. For instance, can the robot in [1] carry a quadrotor in a way similar to that in Fig 8?

4. Overall, I have major concerns about the novelty and impact of this work. In the meantime, I think the experiments are done thoroughly and the figures are of high quality. The robot combines many functions, and it is worthy of publication in a field specific journal.

Minor concerns:

1. In my view, the title is misleading as the authors claim their robot is capable of “collaborative flying”. However, when I review this paper, it looks like the robot is simply carrying a mini-quadrotor to go through a tunnel. I do not think this represents any flight capability. I suggest rephrasing the title.

2. I think the authors shouldn't just highlight the few demonstrations in Table 1 without comparing other performance metrics, such as speed, efficiency, and cost of transport.

[1] Ji, X., Liu, X., Cacucciolo, V., Imboden, M., Civet, Y., El Haitami, A., Cantin, S., Perriard, Y. and Shea, H., 2019. An autonomous untethered fast soft robotic insect driven by low-voltage dielectric elastomer actuators. *Science Robotics*, 4(37), p.eaaz6451.

[2] Gu, G., Zou, J., Zhao, R., Zhao, X. and Zhu, X., 2018. Soft wall-climbing robots. *Science Robotics*, 3(25), p.eaat2874.

Reviewer #3 (Remarks to the Author):

General comments: In this work, one kind of multimodal locomotion ultra-thin soft robots, consisting of dielectric elastomer actuators and electroadhesive pads, have been proposed, which is capable of crawling, climbing and swimming. Several experiments have been conducted to well demonstrate the multimodal locomotion ability and the potential application of exploring narrow spaces. This work can be recommended for publishing in *Nature Communications* after addressing following comments.

Specific comments:

1) In the section of Introduction, although achieving multimodal locomotion is important for narrow spaces, the main challenge to develop dielectric elastomer actuator based multimodal locomotion is unclear. Considering that many dielectric elastomer actuator based robots have been proposed, it is better to explain why multimodal locomotion is challenging.

2) In Fig. 1, many potential applications of the robots have been illustrated. However, those applications are not verified by experiments. It is easy to cause confusion.

3) In the text S1 to S4, analytical models of the TS-DEAs have been proposed. I noticed that the viscoelasticity has been taken into consideration. However, Fig. S5 only shows the static experimental results and verification. Viscoelasticity usually leads to rate-dependent hysteresis effect that has serious influence on the dynamic response of DEA and the control strategy of the DEA-based robots. It is better to further analyze the viscoelasticity of DEAs.

4) The electroadhesive force of the electroadhesive pad usually relies on the material, surface of the substrate. The influence of voltage and substrate should be characterized.

5) The control system is much larger and heavier than that of robots. It may lead to huge challenge for the application in narrow space.

Review Comments

Reviewer #1 (Remarks to the Author):

The authors present an interesting platform for robotics research: the tunable Poisson ratio for the dielectric elastomer actuators is a powerful tool to produce robotic function from design parameters. The results are compelling, as the robots can locomote between varying environments that have been challenging for similar machines in the past.

As it stands the work has three limitations which may prevent acceptance in this journal:

1. the use of VHB elastomer as the working material is disappointing: while the material is commercially available, it has major limitations particularly when it comes to fast actuation, due to its appreciable viscoelasticity. Considering the robotic applications, I urge the authors to consider other materials, including combinations of commercially available Ecoflex 00-30 and Sylgard 184, proven to actuate above 200 Hz. (Citation: Zhao, H., Hussain, A.M., Duduta, M., Vogt, D.M., Wood, R.J. and Clarke, D.R., 2018. Compact dielectric elastomer linear actuators. *Advanced functional materials*, 28(42), p.1804328.). However I understand that changing the elastomer might be reserved for future studies, beyond the scope of the current work.
2. the characterization of the actuators is incomplete: dielectric elastomer actuators are extremely interesting because of the direct conversion of electrical energy into mechanical work. Multiple studies have looked at maximizing the conversion efficiency, one good work can be found here: Bigué, J.P.L. and Plante, J.S., 2011. Experimental study of dielectric elastomer actuator energy conversion efficiency. *IEEE/ASME Transactions on Mechatronics*, 18(1), pp.169-177. The use of the controllable Poisson ratio opens up new possibilities in enhancing the energy conversion. However the energetics of the system are not studied almost at all by the authors. To strengthen this work, and show its clear advantages over state of the art, I encourage the authors to look at how much mechanical work is produced from the input of electrical energy, across ranges of frequency and operation modes. As it stands the robot seems to only be usable in a tethered fashion, so an understanding of energy is needed if it were ever to become autonomous.
3. Operation in the real world needs reliable and resilient robots. Unfortunately, no data is shown on how the performance of these robots changes over time or as a function of actuation cycles. Either would be extremely interesting, and enrich the field of soft robotics.

Reviewer #2 (Remarks to the Author):

In this paper, the authors describe a thin-film robot driven by dielectric elastomer actuators. The robot demonstrates multimodal locomotion such as crawling, climbing, and swimming. Overall, the experimental demonstration is thorough, and the writing and figures are of high quality. However, in my opinion, this paper does not represent significant contributions to the microrobotic and soft robotic

communities and I do not feel it is worthy of publication in Nature Communications. My main concerns and suggestions are described below.

Main concerns:

1. I do not feel the paper has addressed important and fundamental questions in the research fields. First, the performance of the reported robot is not outstanding compared to other DEA-driven robots. The crawling speed is particularly slow compared to the state-of-the-art results [1]. I cannot identify novelty in the DEA fabrication method, and the DEA performance (blocked force, maximum energy/force/displacement output) is not competitive. I do not see how this work can benefit future studies in improving DEA performance. Second, the DEA field is moving towards power autonomy [1], yet this robot is always tethered to an amplifier that commands 2kV- 5kV. Given the high operation voltage and small size, I think that there is a long way for this robot to achieve power autonomy. Third, the reviewer has experience working with aquatic microrobots. One of the biggest challenges is that water is conductive and it has non-negligible surface tension. I believe the authors use silicone oil as the liquid to bypass shorting problems. I do not think this is a good swimming demonstration considering the work has not addressed challenges in preventing shorting and overcoming surface tension. These three big challenges – DEA performance, power autonomy, and high voltage insulation – have not been sufficiently addressed.

2. One novelty of this work focuses on climbing via electrostatic adhesion, but the design is very similar to an existing work [2]. I also do not see significantly improved performance or new physical principles. I think enabling transitioning across a corner represents a new function, but it is incremental as it is not difficult to achieve in the prior robot [2].

3. The Discussion section lacks a detailed description of the work's implication and impact. Why are the chosen demonstrations difficult for existing robots and why are they important? What major bottlenecks have been solved in this work and what is the major novelty? I think the presented robot is still very far from being deployed in realistic applications, and I do not see how these demonstrations are challenging for existing robots. For instance, can the robot in [1] carry a quadrotor in a way similar to that in Fig 8?

4. Overall, I have major concerns about the novelty and impact of this work. In the meantime, I think the experiments are done thoroughly and the figures are of high quality. The robot combines many functions, and it is worthy of publication in a field specific journal.

Minor concerns:

1. In my view, the title is misleading as the authors claim their robot is capable of “collaborative flying”. However, when I review this paper, it looks like the robot is simply carrying a mini-quadrotor to go through a tunnel. I do not think this represents any flight capability. I suggest rephrasing the title.

2. I think the authors shouldn't just highlight the few demonstrations in Table 1 without comparing other performance metrics, such as speed, efficiency, and cost of transport.

[1] Ji, X., Liu, X., Cacucciolo, V., Imboden, M., Civet, Y., El Haitami, A., Cantin, S., Perriard, Y. and Shea, H., 2019. An autonomous untethered fast soft robotic insect driven by low-voltage dielectric elastomer actuators. *Science Robotics*, 4(37), p.eaaz6451.

[2] Gu, G., Zou, J., Zhao, R., Zhao, X. and Zhu, X., 2018. Soft wall-climbing robots. *Science Robotics*, 3(25), p.eaat2874.

Reviewer #3 (Remarks to the Author):

General comments: In this work, one kind of multimodal locomotion ultra-thin soft robots, consisting of dielectric elastomer actuators and electroadhesive pads, have been proposed, which is capable of crawling, climbing and swimming. Several experiments have been conducted to well demonstrate the multimodal locomotion ability and the potential application of exploring narrow spaces. This work can be recommended for publishing in *Nature Communications* after addressing following comments.

Specific comments:

1) In the section of Introduction, although achieving multimodal locomotion is important for narrow spaces, the main challenge to develop dielectric elastomer actuator based multimodal locomotion is unclear. Considering that many dielectric elastomer actuator based robots have been proposed, it is better to explain why multimodal locomotion is challenging.

2) In Fig. 1, many potential applications of the robots have been illustrated. However, those applications are not verified by experiments. It is easy to cause confusion.

3) In the text S1 to S4, analytical models of the TS-DEAs have been proposed. I noticed that the viscoelasticity has been taken into consideration. However, Fig. S5 only shows the static experimental results and verification. Viscoelasticity usually leads to rate-dependent hysteresis effect that has serious influence on the dynamic response of DEA and the control strategy of the DEA-based robots. It is better to further analyze the viscoelasticity of DEAs.

4) The electroadhesive force of the electroadhesive pad usually relies on the material, surface of the substrate. The influence of voltage and substrate should be characterized.

5) The control system is much larger and heavier than that of robots. It may lead to huge challenge for the application in narrow space.

Response to Reviewer 1 (Amendments are marked in green)

General comments:

The Authors present an interesting platform for robotics research: the tunable Poisson ratio for the dielectric elastomer actuators is a powerful tool to produce robotic function from design parameters. The results are compelling, as the robots can locomote between varying environments that have been challenging for similar machines in the past.

Response to general comments:

Thank you for reviewing our paper and for the insightful and constructive comments provided in the follow-up. To address your comments, we have significantly improved our manuscript and added a substantial amount of new experiments, data, and clarifications. In the following, we presented our response to addressing your comments point-by-point. The amendments to the manuscript and supplementary file are marked **in green**.

Specific Comments:

Comment R1-1:

The use of VHB elastomer as the working material is disappointing: while the material is commercially available, it has major limitations particularly when it comes to fast actuation, due to its appreciable viscoelasticity. Considering the robotic applications, I urge the authors to consider other materials, including combinations of commercially available Ecoflex 00-30 and Sylgard 184, proven to actuate above 200 Hz. (Citation: Zhao, H., Hussain, A.M., Duduta, M., Vogt, D.M., Wood, R.J. and Clarke, D.R., 2018. Compact dielectric elastomer linear actuators. *Advanced functional materials*, 28(42), p.1804328.). However I understand that changing the elastomer might be reserved for future studies, beyond the scope of the current work.

Response to Comment R1-1:

Thank you for the constructive comment. We have taken positively your critical comments on the use of VHB. Although time-consuming to explore new silicone-based elastomers, we have implemented the recommended approach (a mixture material of Ecoflex 00-30 and Sylgard 184 in a ratio of 1:1, Sylgard 184 was prepared with a cross-linker ratio of 40:1) for fast actuation supported by new tests and characterisations.

In the revision, we tested the new elastomer (mixture material) on two actuator designs, i.e. Type-C-I and II TS-DEA (Supplementary file Text S1 and Movie S2). The Type-C-I TS-DEA is made of a zero Poisson's ratio tensioning mechanism (stiffness k_T) sandwiched by two actuation layers, while the Type-C-II TS-DEA comprises a zero Poisson's ratio tensioning mechanism (stiffness $2k_T$) sandwiched

by four actuation layers (two on each side). Each actuation layer used in both actuators is identical (e.g., recipe and size). The stiffness of the Type-C-II TS-DEA is two times that of Type-C-I.

We found that using the new elastomer (mixture material), the resonant frequencies of the Type-C-I and II TS-DEAs are 51 Hz and 67 Hz, respectively (Fig.3.L). The resonant frequency of Type-C-II (f_{C-II}) is approximately equal to $\sqrt{n_a}f_{C-I}$, where f_{C-I} is the resonant frequency of Type-C-I, and n_a is the ratio of the number of actuation layers in Type-C-II compared to Type-C-I ($n_a = 2$ in this case). The dynamic maximum displacement at the resonant frequencies for the Type-C-I and II designs with the new elastomer are 0.22mm and 0.13mm (at the electric field strength of 21.7V/um, Fig.3.L), respectively, while the static displacements of both actuators are almost identical at 0.9mm (Fig.3.J). In this case, compared with the Type-C-I actuator, the decrease in the Type-C-II dynamic displacements is mainly because of the damping effect caused by the friction between the two adjacent actuation layers on each side, which are independent of each other in the current design (Supplementary file Text S1). The friction could be eliminated by combining the independent actuation layers into one multiple-layer actuation structure to increase the dynamic displacement, which is an approach that can be implemented in the future (ref-31).

Furthermore, we added directional friction feet to the Type-C-II TS-DEA to test the speed of the Type-C-II TS-Robot (Movie S5). The robot's maximum speed (37mm long) reached 43.1mm/s (1.16 Body length/s), which, by itself, can be considered competitive performance in speed when compared with the published literature. However, we would like to emphasise that the resonant frequency of our proposed design can be further increased by increasing the number of its actuation layers, as it is approximately equal to $\sqrt{n_a}f_{C-I}$. Thus, the speed of the robot could be even higher.

Therefore, one of the novelties of the proposed actuator structure is that its resonant frequency can be tuned by changing the stiffness of the tensioning mechanism (Fig.3L) while not needing to change the elastic modulus and damping ratio of the elastomer by adjusting its recipe. Compared with the state-of-the-art method, the proposed design method is less time-consuming for tuning the resonant frequency of the DE-based actuator, as it does not include adjusting the complex chemical processes of making elastomers.

We have added the corresponding revision to the following parts in the manuscript and the supplementary file.

[Amendments in the revised main manuscript on page 4 - Response to Comment R1-1]

Depending on the requirements of the applications, both VHB-based and silicone-based elastomers can be selected and used in the actuation layer. Further, more importantly, this design enables us to easily tune the resonant frequency of the TS-DEA for fast actuation by mechanically changing the properties (e.g., stiffness) of the tensioning mechanism, which could be less time-consuming to process compared with the widely used state-of-the-art chemical method (31) (Supplementary file Test S1).

[Amendments in the revised main manuscript on pages 7-8 - Response to Comment R1-1]

Here, the EA-Pads offer a great advance, enabling the proposed robots to move on the surfaces of various materials and orientations. However, as the polarisation and depolarisation time of the substrate surfaces by the EA-Pads is 100-300ms per cycle, this bottleneck prevents the proposed TS-Robot from being operated with a high frequency. Hence, we used VHB as the elastomer in the actuation layers for Type-A and -B TS-Robots since it can withstand high electrical field strength to generate large strain and force at low frequencies (1-5Hz).

As a complement, we also tested other designs of TS-Robots for fast actuation by using two types of silicone-based elastomers (31) (one is a mixture material composed of Ecoflex 00-30 and Sylgard 184 in a 1:1 ratio, in which, Sylgard 184 was prepared with a cross-linker ratio of 40:1; the other one is made of pure Ecoflex 0030) and directional friction feet instead of VHB and EA-Pads, respectively.

Type-C TS-Robot: the system contains a zero Poisson's ratio TS-DEA with directional friction feet and actuation layers using silicone-based elastomers. Its actuator is denoted as Type-C TS-DEA (*Supplementary file Text S1*).

- I. Type-C-I TS-DEA: the actuator is made of a zero Poisson's ratio tensioning mechanism (stiffness k_T) sandwiched by two actuation layers (the mixture material).
- II. Type-C-II TS-DEA: the actuator comprises a zero Poisson's ratio tensioning mechanism (stiffness $2k_T$), sandwiched by four actuation layers (the mixture material), two on each side.

[Amendments in the revised main manuscript on pages 8-11 - Response to Comment R1-1]

We investigated the dynamic properties of the Type-A TS-DEA, actuated at the electric field strength of $100\text{V}/\mu\text{m}$ by a square wave linear frequency sweep ranging from 1Hz to 100Hz. Due to the viscoelastic properties of VHB, the displacement of the Type-A TS-DEA decreased to below 0.1 mm when frequencies reached 20Hz. Hence, considering the polarisation and depolarisation time of the substrates by the EA-Pad (less than 10Hz), the demonstrations of the Type-A TS-Robot were primarily conducted at lower frequencies, specifically within the 1-5Hz range.

Fig. 3. Characterisation results of the three types of TS-DEAs. (A) Static displacement of the Type-A TS-DEA vs. Electric field strength; (B) Blocking Force of the Type-A TS-DEA vs. Electric field strength when the actuator is constrained to its initial length; (C) Dynamic displacement of the Type-A TS-DEA vs. Actuation frequency (100V/um); (D) Static bending displacement of the Type-A TS-DEA vs. Electric field strength; (E) Blocking Force of the Type-A TS-DEA bending vs. Electric field strength when the actuator is constrained to its initial shape; (F) Dynamic bending displacement of the Type-A TS-DEA vs. Actuation frequency (100V/um); (G) Static displacement of the Type-B TS-DEA in X and Y axes vs. Electric field strength; (H) Blocking Force of the Type-B TS-DEA vs. Electric field strength when the actuator is constrained to its initial length; (I) Dynamic displacement of the Type-B TS-DEA vs. Actuation frequency (100V/um); (J) Static displacement of the Type-C TS-DEAs vs. Electric field strength; (K) Blocking Force of the Type-C TS-DEAs vs. Electric field strength when the actuator is

constrained to its initial length; (L) Dynamic displacement of the Type-C TS-DEAs vs. Actuation frequency (21.7V/um).

Further, it was found that both Type-C-I and -II DEAs can make a maximum static displacement of 0.9 mm (Fig. 3J) and a blocking force of 0.038N and 0.081N (Fig. 3K), respectively. Then, the dynamics of Type-C-I DEA (tensioning mechanism stiffness k_T , driven by two actuation layers) were tested at an electric field strength of 21.7V/um. We found that its resonant frequency can reach 51 Hz, and its displacement is 0.22mm. In contrast, the resonant frequency of Type-C-II DEA (tensioning mechanism stiffness $2k_T$, driven by four actuation layers) is 67Hz, and its displacement is 0.13mm (Fig. 3L). In this case, compared with the Type-C-I actuator, the decrease in the Type-C-II dynamic displacement is mainly because of the damping effect caused by the friction between the two adjacent actuation layers on each side, which are independent between each other in the current design (*Supplementary file Text S1*). The friction could be eliminated by combining the independent actuation layers into one multiple-layer actuation structure to reduce the damping effect and increase the dynamic displacement, which is an established method in literature that can be implemented in future work (31). More importantly, it can be found that the resonant frequency of the proposed TS-DEAs can be easily tuned by changing the stiffness of the tensioning mechanism while not needing to change the properties of the elastomers by adjusting their recipe.

[Amendments in the revised main manuscript on page 13 - Response to Comment R1-1]

Silicone-based TS-DEAs provide outstanding speed performance due to their capability for high-frequency actuation. We demonstrate the enhanced mobility of the Type-C-II TS-Robot, which is equipped with three-directional friction feet, as detailed in *Supplementary file Text S11*. The robot achieved a maximum speed of 43.1 mm/s, corresponding to 1.16 times its body length per second and 13.1 times its body thickness per second, as shown in Movie S5. The robot was actuated at its resonant frequency (67Hz), which can be further increased by adding more actuation layers, thereby enhancing its speed, as clarified in *Supplementary file Text S1*.

[Amendments in the revised supplementary materials on pages 3-7 - Response to Comment R1-1]

Text S1. Design and working principle of the TS-DEA

Fig. S1A-C presents a schematic detailing the interaction between the actuation layer and the tensioning mechanism in the Y axis throughout the fabrication and actuation phases (the calculation procedure is also applied to the X axis). Initially, the pre-stretched actuation layer is affixed to a tensioning mechanism at its neutral state (length $L_{T,y}$), as depicted in Fig. S1A. Subsequently, they are bonded together and then allowed to relax to their balanced state (length l_y , when unpowered), illustrated in Fig. S1B. Upon application of voltage, the DEA exhibits a voltage-induced displacement, denoted as Δ_y , shown in Fig. S1C.

Fig. S1D illustrates the force-length mapping of the TS-DEA between its actuation layer and tensioning mechanism. The symbol $L_{s,y}$ represents the original length of the actuation layer, which during fabrication is pre-stretched to $L_{T,y}$, aligning with the length of the tensioning mechanism at its neutral state. Once adhered together, the combined structure shrinks to length l_y , achieving equilibrium where the contraction force of all the actuation layers $A\sigma_y$ (A is the cross-sectional area and σ_y is the stress) balances with the tensioning mechanism's extension force $F_{Re,y}$. A voltage-induced force $F_{M,y}$ is generated when a voltage is applied, causing the DEA to extend by a displacement of Δ_y .

Fig. S1 Design and working principle of the TS-DEA. (A) Pre-stretched actuation layer adhered to the tensioning mechanism at its neutral state; (B) TS-DEA in the unpowered state; (C) TS-DEA in the powered state; (D) The force-length mapping of the TS-DEA between its actuation layer and tensioning mechanism.

Fig. S2 illustrates the design and prototypes of Type-C-I TS-DEA, Type-C-II TS-DEA and L-Type-C TS-DEA (The designs and prototypes of Type-A and Type-B TS-DEA are illustrated in Fig. 2 E and F), and their specifications listed in Table. S1.

Fig. S2 Design and prototypes of TS-DEAs. (A) Type-C-I TS-DEA; (B) Type-C-II TS-DEA; (C) L-Type-C TS-DEA. (Note: Type-C-I (two actuation layers) and Type-C-II (four actuation layers) TS-DEA share the same elastomer made of a mixture material; actuation layer material for the L-Type-C TS-DEA (four actuation layers) is made of pure Ecoflex 0030).

Table. S1 Specifications of TS-DEAs

	Type-A TS-DEA	Type-B TS-DEA	Type-C-I TS-DEA	Type-C-II TS-DEA	L-Type-C TS-DEA
Material	VHB 4910	VHB 4910	Mixture ⁽¹⁾	Mixture ⁽¹⁾	Ecoflex 0030
Pre-stretch ratio	4.5*4.5	4.5*4.5	1*1.4	1*1.4	2.5*2.5
Initial elastomer thickness (mm)	1	1	0.1	0.1	0.05
Final elastomer thickness (mm)	0.05	0.05	0.083	0.083	0.01

Dimension (mm) (X*Y*Z)	40*37*1.2	39*39*1.2	40*37*1.2	40*37*1.3	40*37*1.1
Weight (g)	1.2	1.1	1.2	1.4	1.2
Stiffness of actuation layer ⁽²⁾ (N/m)	38.4	Y 38.4 X 38.4	26.0	52.0	27.8
Stiffness of tensioning mechanism ⁽³⁾ (N/m)	273.1	Y 673.3 X 621.3	12.8	23.3	52.0
Resistance ⁽⁴⁾ (MΩ)	1.53	1.22	2.11	3.84	0.41
Capacitance ⁽⁴⁾ (nF)	0.018	0.017	0.067	0.095	1.690

Note:

⁽¹⁾ The mixture material is composed of Ecoflex 00-30 and Sylgard 184 in a ratio of 1:1 (Sylgard 184 was prepared at a cross-linker ratio of 40:1, as referenced in (31)).

⁽²⁾ The value presented here represents the total stiffness of all the actuation layers of the TS-DEA. It is calculated based on the force balance between the tensioning mechanism and the actuation layer, as shown in Fig. S1D. The X and Y means the stiffness of the tensioning mechanism in the X and Y axis.

⁽³⁾ The stiffness of the tensioning mechanism is measured by compressing the mechanism in-plane by 2mm at a speed of 1mm/s and calculating the force-displacement ratio.

⁽⁴⁾ The resistance and capacitance data are measured using the ModuLab XM system, with the values representing averages over frequencies ranging from 50 to 100 Hz.

As illustrated in Fig. S1D, the stiffness of the elastomer and the tensioning mechanism can be calculated by:

$$k_e = \frac{A\sigma_y}{\Delta_y} \quad (S1)$$

$$k_T = \frac{F_{Re-y}}{\Delta_y} \quad (S2)$$

Therefore, the natural frequency of the TS-DEA can be approximated by:

$$f = \frac{1}{2\pi} \sqrt{\frac{k_e + k_T}{m_{DEA}}} \quad (S3)$$

where the m_{DEA} is the effective mass of the TS-DEA.

This equation indicating that the natural frequency of the TS-DEA is adjustable by modifying mechanical parameters, i.e., the actuation layer stiffness k_e and tensioning mechanism stiffness k_T . To substantiate this concept, we propose two types of silicone-based TS-DEAs that utilize a mixed material, namely, Type-C-I TS-DEA and Type-C-II TS-DEA. As illustrated in Fig. S2 A and B, the Type-C-I TS-DEA is constructed with two actuation layers, one on each side of the tensioning mechanism. In contrast, the Type-C-II TS-DEA features four actuation layers, with two on each side,

as detailed in Table. S1 and Eq. (S14) (Illustrated in Supplementary file Text S2). This configuration results in the stiffness of the Type-C-II TS-DEA ($k_e + k_T$) being twice that of Type-C-I TS-DEA.

We found that the resonant frequencies of the Type-C-I and Type-C-II TS-DEAs are 51 Hz and 67 Hz, respectively. The resonant frequency of Type-C-II (f_{C-II}) can be approximated as $\sqrt{n_a}f_{C-I}$, where f_{C-I} is the resonant frequency of Type-C-I and n_a is the ratio of the number of the actuation layers in Type-C-II compared to Type-C-I ($n_a=2$ in this case). Here, we would like to emphasise that the resonant frequency of our proposed design can be increased by increasing the number of its actuation layers.

A significant benefit of DEAs employing silicone materials is their ability to operate at high frequencies, particularly at their natural frequencies where resonance occurs. This design methodology provides a straightforward way to tailor the resonant frequency by adjusting the stiffness of the tensioning mechanism based on specific requirements.

[Amendments in the revised supplementary materials on Movie S2, Movie S5 - Response to Comment R1-1]

Supplementary materials - Movie S2_Characterisation of the Type-C-II TS-DEA

Supplementary materials - Movie S5_Demonstration of Type-C-II TS-Robot with directional friction feet

Comment R1-2:

The characterization of the actuators is incomplete: dielectric elastomer actuators are extremely interesting because of the direct conversion of electrical energy into mechanical work. Multiple studies have looked at maximizing the conversion efficiency, one good work can be found here: Bigué, J.P.L. and Plante, J.S., 2011. Experimental study of dielectric elastomer actuator energy conversion efficiency. IEEE/ASME Transactions on Mechatronics, 18(1), pp.169-177. The use of the controllable Poisson ratio opens up new possibilities in enhancing the energy conversion. However the energetics of the system are not studied almost at all by the authors. To strengthen this work, and show its clear advantages over state of the art, I encourage the authors to look at how much mechanical work is produced from the input of electrical energy, across ranges of frequency and operation modes. As it stands the robot seems to only be usable in a tethered fashion, so an understanding of energy is needed if it were ever to become autonomous.

Response to Comment R1-2:

Thank you for the comment, which inspired us to improve the scientific thoroughness of the work. As per your recommendation, we have added experiments to characterise the energy conversion efficiency of the proposed actuators, including the original designs, the Type A (Poisson's ratio, zero)

and B' (Poisson's ratio, -1) TS-DEAs (using VHB as the elastomer) and the new ones, Type-C-I and -II TS-DEAs (Poisson's ratio, zero, elastomers are made of a mixture material composed of Ecoflex 00-30 and Sylgard 184 in a 1:1 ratio, in which Sylgard 184 was prepared with a cross-linker ratio of 40:1). The actuators were tested across a wide range of frequencies (from zero to their resonant frequencies).

According to the experimental results, the energy conversion efficiency of the Type-A TS-DEA (e.g., 2.26% at 1hz; Poisson's ratio of 0) is higher than that of the Type-B' TS-DEA (Poisson's ratio of -1) in a single direction (e.g., 1.22% at 1Hz), while lower than the total efficiency (e.g., 2.44% at 1Hz) of the Type-B' system by considering its total output. The possible reason is that the elastomer homogenously expands in all directions when a voltage is applied. However, the Type-A design (Poisson's ratio, 0) can remarkably regulate the strains of the elastomer into one direction, which can boost the displacement and efficiency in the specific direction. In contrast, the Type-B' design (Poisson's ratio, -1) enables the strains to make displacement into two directions, which has a lower energy loss than Type-A under a quasi-static condition. *This means the controllable Poisson's ratio design can direct more energy into one desired direction and enhance energy conversion efficiency in the desirable locomotion direction.* However, once a higher frequency is used, the difference in the dynamic responses between the two designs may play a critical role in their efficiencies. For example, the efficiency of the Type-A TS-DEA at 4Hz is higher than the total efficiency of the Type-B' design, as the dynamic displacement of Type-B' decreases faster than that of Type-A with the frequency increases.

Further, we would like to emphasise that we aim to develop robots to explore narrow spaces in high-value industrial equipment, which can be used in a wide range of real-world applications, including, but not limited to, the inspection and maintenance of industrial equipment such as aero-engines, aircraft, and nuclear facilities (see conceptual Fig.1 from which we "mimicked" the applications of our robots). *Due to the high-value nature of these equipment, robots are often required to be tethered for robust communication and safety in these confined spaces to avoid communication failures and retract the robots if they fail.* Otherwise, if a robot fails inside a piece of high-value equipment, e.g., a multimillion-dollar aeroengine, the equipment may need to be disassembled to remove the robot, which can be very time-consuming and costly. Hence, power autonomy is debatable for the proposed TS-Robots in industrial applications. However, we understand the Reviewer would like to see the autonomous potential of the proposed robot.

We have added a discussion in the manuscript to clarify why the proposed robot's tether design is critical for a set of particular industry applications where access through narrow gaps are key features (see details on the *page 26* in the revised manuscript).

[Amendments in the revised main manuscript on page 11 - Response to Comment R1-2]

Based on the characterisation of the proposed TS-DEAs design, we calculated the output power and energy conversion efficiency (40, 41) of these TS-DEAs across a range of frequencies. The calculations, along with the methodology, are detailed in the *Supplementary file Text S8*.

[Amendments in the revised supplementary materials on pages 25-27 - Response to Comment R1-2]

Test S8. Energy conversion calculation of the TS-DEAs

We calculated the energy efficiency of TS-DEAs based the methodology proposed in literature (40, 41). The energy efficiency of a general actuator can be expressed as:

$$\eta = \frac{P_{DEA}}{P_{In}} \quad (S47)$$

where P_{DEA} is the mechanical power of the TS-DEA and P_{In} is the input power (electrical power) of the DEA. Based on the linear assumption, P_{DEA} can be expressed as:

$$P_{DEA} = \frac{1}{2} F_B \Delta f \quad (S48)$$

where F_B and Δ are respectively the block force and voltage-induced displacement of the TS-DEA when actuated at frequency f . P_{In} can be explained as the integration of the time dependent voltage U and current I at frequency f , written as:

$$P_{In} = f \int_0^T UI dt \quad (S49)$$

where T is the period of a single actuation cycle.

We constructed the test setup as shown in Fig. S22 to measure the block force and voltage-induced displacement, therefore, to calculate the mechanical power P_{DEA} . To obtain the P_{In} , U and I were measured simultaneously through the oscilloscope (OSCs), as shown in Fig. S11, while I was indirectly obtained by the voltage value monitored in OSC 1, where the correlation follows:

$$I = \frac{U}{R} \quad (S50)$$

where $R = 1k\Omega$ is the resistance which serially connected to the DEA. U was directly read from OSC 2.

Fig. S11 The schematic of the set up for the measurement of the TS-DEAs input energy.

Based on the characterisation results, we computed the output power and efficiency of the TS-DEAs across various frequency ranges (Table. S3). These ranges included a low frequency (1Hz), the natural frequency of the TS-DEA, and 4Hz, which demonstrated the optimal speed performance for VHB-based TS-Robots.

Type-A and Type-B' TS-DEAs, which share the same actuation voltage due to the identical materials and thickness of their actuation layers, exhibited similar trends of energy conversion efficiency across various frequency ranges. For the Type-A TS-DEA, its output power was 0.45mW at 1Hz and increased to 1.22mW at 4Hz. However, at a higher frequency of 50Hz, the output power of the Type-A TS-DEA decreased to 0.37mW, a drop attributed to the high viscoelasticity of the VHB material, which significantly reduces its displacement. The Type-B' TS-DEA followed a similar pattern, with an output power of 0.50mW at 1Hz, increasing to 1.16mW at 4Hz, and decreasing to 0.79mW at 50Hz. The energy conversion efficiency of both Type-A and Type-B' TS-DEAs decreased with the increase of the frequency due to the reduced output displacement, peaking at 1Hz with efficiencies of 2.26% and 2.44%, respectively.

According to the experimental results, the energy conversion efficiency of the Type-A TS-DEA (e.g., 2.26% at 1hz; Poisson's ratio of 0) is higher than that of the Type-B' TS-DEA (Poisson's ratio of -1) in a single direction (e.g., 1.22% at 1Hz), while lower than the total efficiency (e.g., 2.44% at 1Hz) of the Type-B' system by considering its total output. The possible reason is that the elastomer homogenously expands in all directions when a voltage is applied. However, the Type-A design (Poisson's ratio, 0) can remarkably regulate the strains of the elastomer into one direction, which can boost the displacement and efficiency in the specific direction. In contrast, the Type-B' design (Poisson's ratio, -1) enables the strains to make displacement into two directions, which has a lower energy loss than Type-A under a quasi-static condition. This means the controllable Poisson's ratio design can direct more energy into one desired direction and enhance energy conversion efficiency in the desirable locomotion direction. However, once a higher frequency is used, the difference in the dynamic responses between the two designs may play a critical role in their efficiencies. For example, the efficiency of the Type-A TS-DEA at 4Hz is higher than the total efficiency of the Type-B' design, as the dynamic displacement of Type-B' decreases faster than that of Type-A with the frequency increases.

Further, using the silicone-based material for actuation layers, both Type-C-I and Type-C-II TS-DEA showed an increase in output power with frequency increases, peaking at their resonant frequencies of 0.16mW and 0.22mW, respectively. Their efficiencies followed a similar trend to Type-A and Type-B' TS-DEA, with maximum efficiencies at 1Hz of 1.41% and 1.81%, respectively.

Table. S3 Mechanical power and efficiency of four types of TS-DEAs

	Type-A TS-DEA	Type-B' TS-DEA ⁽²⁾	Type-C-I TS-DEA	Type-C-II TS-DEA
Actuation voltage (V)	5000	5000	1800	1800

Mechanical power	1Hz	0.45mW	0.50mW	0.01mW	0.02mW
	4Hz	1.22mW	1.16mW	0.03mW	0.08mW
	High frequency ⁽¹⁾	0.37mW (50Hz)	0.79mW (50Hz)	0.16mW (51Hz Natural frequency)	0.22mW (67Hz Natural frequency)
Efficiency	1Hz	2.26%	2.44%	1.41%	1.82%
	4Hz	1.58%	1.40%	1.11%	1.53%
	High frequency	0.04% (50Hz)	0.07% (50Hz)	0.57% (51Hz Natural frequency)	0.31% (67Hz Natural frequency)

Note:

⁽¹⁾ No resonant frequency was detected in Type-A and Type-B' TS-DEA during the sweep frequency experiment due to the high viscoelasticity of the VHB material. Consequently, 50Hz, representing a high frequency (closed to the resonant frequency of the Type-C TS-DEAs), was selected to calculate their power and efficiency for comparison with the Type-C TS-DEAs.

⁽²⁾ To enable comparisons for assessing how the controllable Poisson's ratios (0 and -1) of the tensioning mechanisms affect the TS-DEA output power and efficiency, the VHB-based Type-A and Type-B' TS-DEAs are included in this table. The parameters for these tensioning mechanisms are derived from Fig. 2B and Eq. (S12) where the key parameters, e.g., width of the diagonal struts w (0.65mm), are identical. We also calculated the power and efficiency for the Type-B TS-DEA (width of the diagonal struts w , 0.50mm): the power outputs are 0.43mW at 1Hz, 1.39mW at 4Hz, and 0.60mW at 50Hz, while the corresponding efficiencies are 2.10%, 1.70%, and 0.05%, respectively.

The data calculated for the Type-B/B' TS-DEA represented the combined sum of the X and Y axes.

Comment R1-3:

Operation in the real world needs reliable and resilient robots. Unfortunately, no data is shown on how the performance of these robots changes over time or as a function of actuation cycles. Either would be extremely interesting and enrich the field of soft robotics.

Response to Comment R1-3:

Thank you for the comment. We have carried out additional experiments to characterise the performance changes of the proposed actuators over time. We characterised the performance of two TS-DEAs, including Type-A (VHB-based) and Type-C-I (silicone-based) TS-DEAs, over a period of two weeks. We have added the corresponding revision to the Supplementary file Text S9.

[Amendments in the revised main manuscript on page 11 - Response to Comment R1-3]

Further, we conducted ageing tests on the Type-A and Type-C-I TS-DEAs to evaluate their performance changes over a period of fifteen days. The Type-A TS-DEA, utilising VHB as the elastomer material, is known for its high viscoelastic properties and showed a gradual and prolonged reduction in displacement in the first ten days before stabilising. In contrast, the Type-C-I TS-DEA, made from a silicone-based material, demonstrated a quicker reduction in displacement within the first five days and then stabilised, as detailed in *Supplementary file Text S9*.

[Amendments in the revised supplementary materials on pages 28-29 - Response to Comment R1-3]

Text S9. Ageing test of Type-A TS-DEA and Type-C-I TS-DEA

We conducted the ageing tests on the Type-A (VHB-Based) and Type-C-I (Silicon-Based) TS-DEAs to evaluate their performance changes over a period of fifteen days. During the tests, we measured the displacement and blocking force of the actuators at varying frequencies. These frequencies were chosen to represent different performance capabilities, including 1Hz for testing the robot's low-frequency performance, 4Hz as the frequency of the Type-A TS-DEA for the locomotion demonstrations and 51Hz, the natural frequency of the Type-C-I TS-DEA for its locomotion test. We used a square wave to actuate the TS-DEAs and recorded the peak-to-peak values of the displacement and blocking force at the fifth cycle, at which point the performance of the DEAs is stabilised (Due to the creeping effect of the DEA, the peak-to-peak value is stabilised after five cycle of actuation).

The experimental results reveal that the displacement of both the Type-A and Type-C-I TS-DEAs exhibits exponential decay over time. The Type-A TS-DEA, utilising VHB as the elastomer material known for its high viscoelastic properties, shows a gradual and prolonged reduction in displacement. For example, at 1Hz, the displacement of the actuator decreased from 0.96 mm to 0.87 mm over ten days and then stabilised at 0.87 mm, as shown in Fig. S12 A. In contrast, the Type-C-I TS-DEA, made from a silicone-based material, demonstrated a quicker reduction in displacement, falling from 0.56 mm to 0.44 mm within the first 5 days before stabilising at 0.44 mm, as illustrated in Fig. S12 C. Notably, no significant reduction in force performance was observed for either Type-A or Type-C-I TS-DEA, as depicted in Fig. S12 B and D.

Fig. S12 Ageing test results for the Type-A TS- and Type-C-I TS-DEAs. (A) Displacement versus times for the Type-A TS-DEA; (B) Blocking force versus time for the Type-A TS-DEA; (C) Displacement versus time for the Type-C-I TS-DEA; and (D) Blocking force versus time for the Type-C-I TS-DEA. (The dashed lines represent the exponential decay fitting curves based on the experimental data)

Response to Reviewer 2 (Amendments are marked in blue)

General comments:

In this paper, the authors describe a thin-film robot driven by dielectric elastomer actuators. The robot demonstrates multimodal locomotion such as crawling, climbing, and swimming. Overall, the experimental demonstration is thorough, and the writing and figures are of high quality. However, in my opinion, this paper does not represent significant contributions to the microrobotic and soft robotic communities and I do not feel it is worthy of publication in Nature Communications. My main concerns and suggestions are described below.

Response to general comments:

Thank you for taking the time to review our paper and for the critical comments that motivated us to improve the submission. To address your comments and concerns, we have added a substantial number of new experiments, data, modelling and explanations to improve the manuscript. In the following part of the document, we presented our response to addressing your comments point-by-point. We hope that our efforts to improve the manuscript significantly will be positively observed. Kindly note that amendments to the main manuscript and supplementary materials related to your comments are marked **in blue**.

Specific Comments

Comment R2-1.

I do not feel the paper has addressed important and fundamental questions in the research fields. First, the performance of the reported robot is not outstanding compared to other DEA-driven robots. The crawling speed is particularly slow compared to the state-of-the-art results [1]. I cannot identify novelty in the DEA fabrication method, and the DEA performance (blocked force, maximum energy/force/displacement output) is not competitive. I do not see how this work can benefit future studies in improving DEA performance.

Response to Comment R2-1:

Thank you for the comment. Firstly, we would like to emphasise that our Thin Soft robots (TS-Robots) distinguish themselves by being ultra-thin, i.e. *near-millimetre (Type-A TS-Robot, 1.7mm thickness) and submillimetre-thickness (C-Type-A TS-Robot, 0.8mm thickness)*, which enable access to a class of ultra-narrow environments encountered in industrial and civil industries. Indeed, other DEA-driven robots have outstanding performances in free spaces, but none of them can go through *near-millimetre and submillimetre-high* narrow gaps. Then, as will be noted later, our TS-Robots (Type-A and Type-C-II TS-Robot) have outstanding performances in speed and payload/self-weight ratio, compared with the state-of-the-art, when the state-of-art and our robots are working in the same narrow-gap working conditions. (Supplementary file Fig. S29)

Please note that narrow gaps are often found in industry, such as the air gap between the stator and rotor in generators/motors (generally around 1mm; the electrical power systems will play a pivoting position in the Net-Zero future). However, no existing solutions can access such narrow spaces, including off-the-shelf borescopes (mini 4mm in diameter) and rigid or soft robots. We have added a demonstration to prove the concept of using the proposed TS-Robot to access the air gap between the stator and rotor in the generators of electrical-powered aero-engines (1.2mm high) for inspection.

Secondly, we would like to clarify the novelty of this work relies on the following four aspects:

1. Novelty 1 - To the authors' best knowledge, the proposed system is the thinnest DEA-actuated robot with two locomotion gaits (e.g., C-Type-A TS-Robot, 0.8mm thickness; Type-A TS-Robot, 1.7mm thickness). It enables the possibility to access the previously inaccessible near-millimetre and even submillimetre gaps (Supplementary file Text S10) and locomotes in multiple domains (e.g., solid and liquid), which no other DEA-based robots can achieve.
2. Novelty 2 - The proposed TS-Robot can achieve a higher payload/self-weight ratio (29:1) compared with the state-of-the-art DEA-driven robot (less than 10:1) ([ref 2, provided by the Reviewer]); this critical feature enables the proposed robots to carry another robot (“twin” TS-Robots or mini-drone) for collaborative locomotion and manipulation, such as the transition between surfaces and gaps (Fig.7C and Fig8.C and D).
3. Novelty 3: In the revised manuscript, we introduced a new method for tuning dielectric elastomer actuators’ resonant frequency by mechanically adjusting the tensioning mechanism's stiffness. Compared with the state-of-the-art method (a chemical approach, *ref-31*), this mechanical method is less time-consuming for turning the DEA properties to achieve the targeted resonant frequency.
This is a fundamentally new method for improving DEA performance, which can be applied to a wide range of DEA designs. We have added supplement experiments to demonstrate this claim; the details can be found in “Response to Comment 1-1”.
4. Novelty 4 - The proposed TS-Robot can be driven at high speed by using directional friction feet. In the revised manuscript, to improve the speed performance, we used a silicone-based elastomer that can be substituted for VHB (the elastomer used in the initial submission). The maximum speed of the new-built Type-C TS-Robot is 43.1mm/s (Body Length 37mm), which is 1.16 times Body Length/s (faster than the state-of-the-art [ref 1,2, provided by the Reviewer]). Further, to make a fair comparison between the state-of-art and our robots, we also compared the speed with the robot thickness (Body Thickness/s). It can be found that the speed of our robot is 13.06 times body thickness/s, which is the highest speed compared with the state-of-art (Supplementary file Fig. S29).

[Amendments in the revised main manuscript on page 23-24 - Response to Comment R2-1]

Thanks to the proposed structure of the TS-DEA, these robots can be built with near-millimetre and submillimetre thickness, and achieve two types of gaits, i.e., a linear in-plane elongating/shortening and an out-of-plane bending gait (Challenge One). Based on this design, the TS-Robots with thin profiles can independently cross various obstacles (e.g., narrow gaps, walls, liquid pools) while operating in solid and liquid domains or transitioning between them. We demonstrated that a single Type-A TS-Robot performs multimodal locomotion, including crawling, climbing, steering, swimming, and landing, for exploring environments containing multiple types of obstacles and even multiple domains. Further, the TS-DEA can be designed with more DOFs and anisotropy mobility in different axes by tuning the Poisson's ratio of the tensioning mechanism (e.g., Type-B TS-DEA).

More importantly, the TS-Robots can also be joined among themselves to work together for transitioning between surfaces (e.g., wall and floor) and operate as manipulators (e.g., repairing circuits and sending programmable signals), because of their high payload/self-weight ratio (maximum 29:1) and stiffness (311.5N/m and 4.5N/m for linear and bending movements) (Challenge Two). In addition, we demonstrated that the TS-Robots can collaborate with other types of robots as "carriers", for example, by carrying a miniaturised drone with a TS-Robot, enabling both the "carrier" and "passenger" with extra locomotion mode (e.g., Drone – flying over deep trenches; TS-Robot: pass through difficult-to-access spaces).

Further, the proposed design enables a new method for tuning dielectric elastomer actuators' resonant frequency by mechanically adjusting the tensioning mechanism's stiffness. Compared with the state-of-the-art method (a chemical approach, (31)), this mechanical method could be less time-consuming for turning the DEA properties to achieve the targeted resonant frequency. This is a fundamentally new method for improving DEA performance, which can be applied to a wide range of DEA designs. The Type-C TS-Robot has demonstrated outstanding performance in speed, reaching 43.1mm/s (Body Length 37mm, Body Thickness 3.3mm), which is 1.16 times Body Length/s and 13.06 times Body Thickness/s.

Comment R2-2.

Second, the DEA field is moving towards power autonomy [1], yet this robot is always tethered to an amplifier that commands 2kV- 5kV. Given the high operation voltage and small size, I think that there is a long way for this robot to achieve power autonomy.

Response to Comment R2-2:

Thank you for the comment. We agree with the Reviewer that power autonomy is important for the DEA field. We have built a new version of TS-Robots driven at low voltage, i.e. 220V, here named Low-Voltage-Type-C TS-Robot (L-Type-C), to demonstrate its potential for power autonomy. We added a brief discussion of the design, manufacturing, and performance of L-Type-C TS-Robot in the revised manuscript (page 14), Supplementary file Text S11 and Movie S6.

Further, we would like to emphasise that this paper aims to develop near and submillimetre-thickness robots to explore narrow spaces in high-value industrial equipment, which can be used for a wide range of real-world applications, including, but not limited to, the inspection and maintenance of industrial equipment such as aero-engines, aircraft, and nuclear facilities. *Due to the high-value nature of these equipment, robots are often required to be tethered for robust communication and safety in these confined spaces to avoid communication failures and enable their retrieval in case of any form of failure.* Otherwise, if a robot fails inside high-value equipment, e.g., a multimillion-dollar aeroengine, the equipment may need to be disassembled to remove the robot, which can be highly time-consuming and costly. Hence, power autonomy is debatable for the proposed TS-Robots in industrial applications, although it could be desirable in other applications. We have added a discussion on page 26 in the revised manuscript to clarify why the proposed robot's tethered design is critical for industry applications.

Further, as a real-world application example of our TS-Robots, we have added a demonstration of using the proposed TS-Robot to inspect the air gap between the stator and rotor in the generators of electrical aero-engines (1.2mm high). The demonstration details can be found on page 12-13 of the revised manuscript file, the revised Supplementary file Text S10 and Movie S4.

[Amendments in the revised main manuscript on page 8 - Response to Comment R2-2]

Low-Voltage-Type-C TS-DEA (L-Type-C TS-Robot): the actuator comprises a zero Poisson ratio tensioning mechanism, sandwiched by four actuation layers (pure Ecoflex 0030; two on each side; each layer is 10um in thickness), for low-voltage actuation.

[Amendments in the revised main manuscript on pages 12-14 - Response to Comment R2-2]

Fig. 4. Locomotion test of the TS-Robots (Poisson's ratio, zero). (F) Diagram of C-Type-A TS-Robot for inspection of the 2.5MW generator; (G) Prototype of L-Type-C TS-Robot.

As an example of real-world applications of the proposed TS-Robots, we proved the concept of using a TS-Robot to inspect electrical generators for Rolls-Royce electrical aeroengine (Fig. 4F). In this demonstration, a variant of the Type-A TS-Robot (Curved-Type-A TS-Robot, denoted as C-Type-A TS-

Robot, 0.8mm thickness) was tested to evaluate the feasibility of in-situ inspections of the magnetic field degradation of an electrical generator for aircraft propulsion (i.e. an actual trend to reduce CO2 emission globally), through the air gap between its rotor and stator. The motivation for the inspection is that the electrical generator for electrical-powered aero-engines (e.g., the 2.5MW generator for the Rolls-Royce AE2100 hybrid power system) features a rotor with permanent magnets, and the properties of these magnets degrade over time, necessitating regular in-situ surveys to ensure the safety and efficiency of the generator with minimal financial and time cost. However, the 1.2mm-high air gap between the rotor and stator renders it inaccessible for conventional tools, including rigid and soft robots. Here, we demonstrated the C-Type-A TS-Robot (Supplementary file Text S10) for locomotion in the narrow air gap of an electrical generator prototype of Rolls-Royce AE2100 hybrid power system mock-up (Movie S4). Through this demonstration, we show the TS-Robot could be used to address the challenges of accessing extremely narrow gaps in high-value industrial systems.

Further, we tested a new version of one-DoF TS-Robots that operates at a low voltage of 220V, named Low-Voltage-Type-C TS-Robot (L-Type-C TS-Robot, Fig. 4G), to showcase its potential for power autonomy. Actuated at its resonant frequency (86Hz), the robot reached a maximum crawling speed of 12.4 mm/s, equivalent to 0.33 times its body length per second and 3.75 times its body thickness per second (Movie S6). We also demonstrate the load-carrying capacity of the L-Type-C TS-Robot, which can carry up to 3g (2.5 times its self-weight), with results presented in Supplementary file Table S4.

[Amendments in the revised main manuscript on page 26 - Response to Comment R2-2]

Further, the TS-robots were proposed to explore narrow spaces in high-value industrial equipment/environments, which can be used in a wide range of real-world applications, including, but not limited to, the inspection and maintenance of industrial equipment such as aero-engines, aircraft, and nuclear facilities. These applications often require the robots to be tethered for robust communication and safety in these confined spaces to avoid communication failures and retract the robots if they fail. Hence, the current robot is powered and controlled through thin wires. However, it may limit the reach range of the robots due to the constraints of the wire length, which can be addressed by using a battery-powered untethered design containing ultra-thin-layer DEAs.

[Amendments in the revised supplementary materials on pages 30-34 - Response to Comment R2-2]

Test S10. Demonstration of the C-Type-A TS-Robot (Curved Type-A TS-Robot) for in-situ inspection of the 2.5MW generator prototype through a 1.2mm-high air gap

As an example of real-world applications of the proposed TS-Robots, we proved the concept of using a TS-Robot to inspect electrical generators for Rolls-Royce electrical aeroengine. In this

demonstration, the TS-Robot was deployed within the air gap (1.2mm-high) between its rotor and stator of an electrical generator mock-up.

The reason for the inspection is that the electrical generator for electrical-powered aero-engines (e.g., the 2.5MW generator for the Rolls-Royce AE2100 hybrid power system) features a rotor with permanent magnets, and the properties of these magnets degrade over time. This requires regular in-situ surveys to ensure the safety and efficiency of the generator with minimal cost and time. However, the 1.2mm-high air gap between the rotor and stator renders it inaccessible for conventional tools, including rigid and soft robots.

Specifications of the 2.5MW generator for the Rolls-Royce AE2100 hybrid power system (a mock-up of the electrical generator)

Fig. S13A presents a Rolls-Royce hybrid power system with an electrical generator, which has a 1.2mm air gap between the stator and rotor. The material of the stator is metallic coil, and the material of the rotor is magnet. Hence, it is difficult to use the widely used small-scale magnetic-driven thin robots in the medical field, as the materials of the generator can affect the magnetic field used for controlling the robot.

To showcase the capability of our TS-Robot to access this confined space, we constructed a mock-up of the electrical generator (Fig. S13B-D), which shares identical dimensions with the real generator. This mock-up, with a 1.2mm-high gap between the stator and rotor, was designed to demonstrate the robot's ability to access and move within such tight confined spaces. To enable the observation of the robot locomotion in the mock-up, a transparent material, PET, was selected as the casing of the mock-up, upon which the robot crawls. Further, a series of magnetic rings from a real electrical generator (i.e., the Rolls-Royce AE2100 hybrid power system) was used to build up the rotor of the mock-up.

Fig. S13 Specifications of the mock-up of a Rolls-Royce hybrid power system with an electrical generator. (A) Illustration of a Rolls-Royce hybrid power system with an electrical generator; (B) A mock-up of the electrical generator; (C) Diagram of the C-Type-A TS-Robot accessing the electrical generator; (D) Front view of the air gap between the stator and rotor.

Design and fabrication of the C-Type-A TS-Robot

A variant of the Type-A TS-Robot (Curved-Type-A TS-Robot, denoted as C-Type-A TS-Robot, 0.8mm thickness) was designed to access the electrical generator mock-up, which is slightly different from the Type-A TS-Robot in four key aspects:

Firstly, it's important to highlight that the air gap is circular in shape. Consequently, the body of the C-Type-A TS-Robot was designed with a curved shape, matching the radius of 148mm to align with

the air gap's curvature, as shown in Fig. S14A. The curved body incorporates a pre-curved tensioning mechanism, initially shaped to a 148 mm radius using the hot stamping method and subsequently cut into the desired pattern with a laser cutter, the same as the method for making the tensioning mechanism of the Type-A TS-DEA.

Fig. S14. Design of the Curved-Type-A TS-Robot (C-Type-A TS-Robot, Radius - 148mm). (A) Prototype of the C-Type-A TS-Robot; (B) CAD model of the C-Type-A TS-DEA; (C) Exploded view of the C-Type-A TS-DEA.

Secondly, to protect the electrode and insulate it from the external environment, the structure of the TS-DEA was modified, as depicted in Fig. S14B and C. Two pre-curved tensioning mechanisms are used to sandwich the actuation layer, enhancing both structural integrity and insulation.

Thirdly, the thickness of the C-Type-A TS-Robot was reduced to 0.8mm to fit within a 1.2mm air gap. This reduction was achieved by minimising the tensioning mechanism's thickness to 0.25mm, optimising the robot's profile to suit its operational environment.

Finally, to enable the C-Type-A TS-Robot to crawl at all angles around the air gap—horizontally, vertically, and inversely—three EA-Pads and two actuation sections were incorporated when designing the robot (Fig. S14A). This configuration provides sufficient adhesive force between the robot and the substrate, ensuring secure locomotion.

Demonstration of the C-Type-A TS-Robot inside the air gap of an electrical generator mock-up

We demonstrated the C-Type-A TS-Robot's capability for locomotion using the same gait as that of the SK-TS-Robot, Gait one, as depicted in Fig. S26B. In Movie S4, we showcased the C-Type-A TS-Robot accessing the air gap. Two cases were presented: in the first, the robot is shown crawling horizontally within the air gap for a distance of 100mm in 112 seconds, starting from the entrance (Fig. S15A); the second case features the robot crawling inverted on the stator inside the air gap (Fig. S15B).

Fig. S15. Demonstration of the C-Type-A TS-Robot in the airgap mock-up. (A) C-Type-A TS-Robot crawling horizontally to access the air gap; (B) C-Type-A TS-Robot crawling inverted to access the air gap.

Test S11. Demonstration of TS-DEA crawling using the directional friction feet

The TS-DEA employing silicone-based elastomer (one is a mixture material composed of Ecoflex 00-30 and Sylgard 184 in a 1:1 ratio, in which Sylgard 184 was prepared with a cross-linker ratio of 40:1; the other one is made of pure Ecoflex 0030) exhibits higher actuation frequencies compared to the Type-A and Type-B TS-DEAs. Therefore, we demonstrated the enhanced mobility of the Type-C-II TS-Robot and L-Type-C TS-Robot, both equipped with three-directional friction feet, as illustrated in Fig. S16A. The directional friction feet are fabricated from 0.1 mm thick PET sheets, which are cut using a laser cutter and then folded into the shape depicted in Fig. S16B. Each foot measures 4 mm in width and 2 mm in height. These feet are glued to the front and back edges of the TS-Robot. The robot's capability for horizontal crawling has been demonstrated on a white high-gloss acrylic sheet.

Fig. S16 Diagram of TS-Robot with the directional friction feet. (A) CAD model of Type-C-II TS-Robot; (B) Three directional frictional feet attach to TS-DEA. Note: L-Type-C TS-Robot was designed in an identical format, but with a different material of the elastomer (mixture material for Type-C-II TS-Robot and pure Ecoflex 0030 for L-Type-C TS-Robot).

From the analysis of the Type-C-II TS-DEA, we applied a 1800V, 67 Hz (natural frequency of the Type-C-II TS-DEA) step wave to actuate the Type-C-II TS-Robot. The robot achieved a maximum speed of 43.1 mm/s, which corresponds to 1.16 times its body length per second and 13.1 times its body thickness per second.

For the L-Type-C TS-Robot, its actuation was achieved using a 220V, 86 Hz (natural frequency of the L-Type-C TS-DEA) step wave, reaching a maximum crawling speed of 12.4 mm/s, equivalent to 0.33 body length per second and 3.75 body thickness per second. Additionally, we demonstrated the load-carrying capability of the L-Type-C TS-Robot, with results presented in Table. S4 and Movie S6.

Table. S4 L-Type-C TS-Robot speed versus payload.

Payload (g)	Speed (mm/s)	Speed (Body Length/s)	Speed (Body Thickness/s)
0	12.36	0.33	3.75
0.25	4.79	0.13	1.45
0.5	4.68	0.13	1.42
1	3.97	0.11	1.20
1.5	2.21	0.06	0.67
1.75	2.06	0.06	0.62
2	1.9	0.05	0.58
2.5	1.8	0.05	0.55
3	1.1	0.03	0.33

[Amendments in the revised supplementary materials on **Movie S4 and Movie S6** - Response to Comment R2-2]

Supplementary materials - Movie S4_Demonstration of C-Type-A TS-Robot in the 2.5MW generator mock-up

Supplementary materials - Movie S6_Demonstration of L-Type-C TS-Robot with directional friction feet

Comment R2-3.

Third, the reviewer has experience working with aquatic microrobots. One of the biggest challenges is that water is conductive and it has non-negligible surface tension. I believe the authors use silicone oil as the liquid to bypass shorting problems. I do not think this is a good swimming demonstration considering the work has not addressed challenges in preventing shorting and overcoming surface tension. These three big challenges – DEA performance, power autonomy, and high voltage insulation – have not been sufficiently addressed.

Response to Comment R2-3:

Thank you for the comment. Kindly note that in the initial submission, we demonstrated the proposed TS-Robot swimming in the water; proving our robot design addresses the short-circuit problems in conductive liquid (Movie S8). In the design of our TS-Robots, the positive poles of the two actuation layer electrodes are sealed within a closed space formed by the elastomer membranes, and the negative poles are exposed to the water (Fig. S17D and E in the revised supplementary file Text S12). Here, water serves as the conductor of the electric current to the negative poles, leveraging its inherent electrical conductivity.

Furthermore, without losing generality, we did new experiments included in the revised paper to test the TS-Robot on which the connections of the negative pole to the DEA are hard-wired to the power supply (Fig. S17A and B in the revised supplementary file Text S12). Hence, in this new demonstration,

we did not use the conductivity of the liquid, i.e. water. As such, it proves our TS-Robot can swim in any liquids regardless of their electrical conductivity.

Finally, in the initial submission, we demonstrated the robot crawling into as well as submerged in the liquid media, then crawling out from the submerged condition, which proves the robot can overcome the surface tension of the liquid for cross-domain locomotion. The details can be found in Movie S9 and Fig.6 (C) and (D) in the main manuscript.

Hence, our TS-Robots can cope with fully liquid (conductive and uncondusive) and hybrid solid-liquid environments.

[Amendments in the revised supplementary materials on Pages 37-38 - Response to Comment R2-3]

Text S12. Type-A TS-Robot swimming in water

In the design of the Type-A TS-Robot, the electrodes of the actuation layers are directly exposed to the air/the non-conductive liquids (e.g., oil), when it swims (Movie S8). For swimming in conductive liquids, we proposed two design methods for the wire connection and insulation (water) to avoid shorting problems:

- Three-pole wiring method (Fig. S17 A-C): The inner side electrodes of Actuation layers 1 and 2 are individually connected to a high voltage (HV) positive pin, while the outer electrodes of the two actuation layers, exposed to the air, share the same connection to the HV negative pin.
- Two-pole wiring method (Fig. S17 D-F): Similar to the first method, the two inner side electrodes are individually connected to an HV positive pin. However, the HV negative pin is exposed to water and connected to the two outer electrodes via the water. Here, the conductive water carries the current as the connection between the negative poles of the power supply and electrodes.

For both wiring methods, the two inner electrodes are sealed within a closed space formed by the elastomer membrane. The connections between the electrodes and the wires are sealed using a silicone-based acid sealant, with an encapsulation layer thickness of approximately 1mm.

Fig. S17 Working principle of two wiring methods of Type-A TS-Robot for swimming in conductive media (water). (A) Three-pole wiring method, Actuation layer 1 is powered; (B) Three-pole wiring method, Actuation layer 2 is powered; (C) Swimming demonstration of the Type-A TS-Robot using the three-pole wiring method in water; (D) Two-pole wiring method, Actuation layer 1 is powered; (E) Actuation layer 2 is powered; (F) Swimming demonstration of the Type-A TS-Robot using the two-pole wiring method in water.

[Amendments in the revised supplementary materials on Movie S8 - Response to Comment R2-3]

Supplementary materials - Movie S8_Swimming test of Type-A TS-Robot in liquid (Silicone oil and Water)

Comment R2-4:

One novelty of this work focuses on climbing via electrostatic adhesion, but the design is very similar to an existing work [2]. I also do not see significantly improved performance or new physical principles. I think enabling transitioning across a corner represents a new function, but it is incremental as it is not difficult to achieve in the prior robot [2].

[2] Gu, G., Zou, J., Zhao, R., Zhao, X. and Zhu, X., 2018. Soft wall-climbing robots. *Science Robotics*, 3(25), p.eaat2874.

Response to Comment R2-4:

Thank you for the comment. We would like to clarify that making the transition between horizontal and vertical surfaces would present some significant challenges for the state-of-the-art prior robot [2]; this requires the robot to be capable of lifting the front feet onto the vertical surface. Based on the design of the prior robot [ref-2] provided by the reviewer, the saddle-shape robot was designed to push its front

feet slide on the surface for crawling locomotion (Fig3A in the prior work [2], Fig. R1), which could not lift its front feet to make such transition.

Fig. R1 Locomotion principle of the saddle-shape robot (Fig3A in the prior work [2])

However, the prior saddle shape robot [2] *may be* able to make the transitions by two-robot collaboration, which requires one robot pushing another one onto the vertical surface and then the first robot pulling the second one onto the vertical surface, but we have not seen any collaboration between the prior saddle shape robot reported.

We have demonstrated such collaboration between two TS-Robots to make the transitions between vertical and horizontal surfaces (Fig. 7 C and G). The enabling factors of the transition by two-robot collaboration are the high payload/self-weight ratio (29:1) of our robot and the high stiffness of its body in its length direction (311.5N/m), which enables the robot to be pushed up to the vertical surface without bulking. In comparison, the saddle shape robot [2] has a payload/self-weight ratio (10:1, estimated), while its body has low stiffness in the length direction (17.5N/m, estimated) to enable displacement for crawling. For the saddle shape robot [2] to perform a similar two-robot collaboration for transitions between surfaces, it may cause the first robot to be compressed like a spring between the vertical surface and the second robot, instead of going up to the wall. Hence, performing transitions across a corner could be a challenge for the prior robot [2] (see the following schematic, Fig. R2).

Fig. R2 Authors' consideration of the challenges in using the saddle-shape robot (as per design of [2], provided by the Reviewer) when trying to collaborate for transitioning between horizontal and vertical surfaces. (A) Configuration of two saddle-shape robot collaboration; (B) The left robot, due to the low stiffness of its saddle-shaped structure, tends to be compressed and get stuck at corners instead of ascending the wall.

As such, we would like to emphasise that, because of our novel design, *our robots can make transitions across corners in the environment with near and submillimetre-high narrow entrances, which the existing robots cannot access*. This is a difference between our robot and the state-of-the-art works. (Find details in Response to Comment R2-1)

Comment R2-5:

The Discussion section lacks a detailed description of the work's implication and impact. Why are the chosen demonstrations difficult for existing robots and why are they important? What major bottlenecks have been solved in this work and what is the major novelty? I think the presented robot is still very far from being deployed in realistic applications, and I do not see how these demonstrations are challenging for existing robots. For instance, can the robot in [1] carry a quadrotor in a way similar to that in Fig 8?

[1] Ji, X., Liu, X., Cacucciolo, V., Imboden, M., Civet, Y., El Haitami, A., Cantin, S., Perriard, Y. and Shea, H., 2019. An autonomous untethered fast soft robotic insect driven by low-voltage dielectric elastomer actuators. *Science Robotics*, 4(37), p.eaaz6451.

Response to Comment R2-5:

Thank you for the comment.

(The Discussion section lacks a detailed description of the work's implication and impact.....I think the presented robot is still very far from being deployed in realistic applications).

The primary challenge for the applications targeted by the proposed robots in this paper is navigating narrow gaps in the environment, particularly near-millimeter and submillimeter-high gaps, that require exploration. Such narrow gaps are often found at places that need inspection and maintenance in industry, but there are currently no solutions for accessing them. *One real-world application example, including its commercial reasons and demonstration, i.e. navigation within the 1.2mm gap between the stator and rotor of an electrical generator for future electrical-powered aeroengine with details, is given in "Response to Comment R2-2".*

Other barriers, such as conductive or nonconductive liquids (see Fig. 6 and Movie S8, S9– hybrid solid-liquid environment), surfaces with varying orientations (see Fig. 5 and Movie S7), transitions between horizontal and vertical surfaces (see Fig. 7 and Movie S10, S11), and deep gaps/trench (see Fig. 8 and Movie S12), are the secondary challenges (but not negligible) in these environments. There might be existing solutions for addressing these secondary challenges in *free space*, but none can be used to access near-millimetre and submillimetre-high narrow gaps, which are the primary barriers in industrial environments.

In this paper, we developed a class of Thin-soft robots that can address both the primary and secondary challenges, which is the bottleneck for a class of industrial scenarios. According to the literature (Table 1 in the main manuscript), our robots are the only ones that can handle both challenges at once for industrial environments. Further, our robots have outstanding performance in speed and payload/self-weight ratio when they are tested in the same condition (under narrow gaps) as the state-of-the-art (Response to Comment R2-1 and Response to Comment R2-7).

Thank you for the comment - (I do not see how these demonstrations are challenging for existing robots. For instance, can the robot in [1] carry a quadrotor in a way similar to that in Fig 8?)

Based on the specifications of the robot in reference [1] provided by the Reviewer, it could be very challenging for it to carry a quadrotor. For carrying the off-the-shelf quadrotor (36g, one of the lightest off-the-shelf quadrotors), our Type-A TS-Robot (1.4g) demonstrated its payload/self-weight ratio reaches 29:1. However, the payload/self-weight ratio of the robot in reference [1] is 1:1, and the max payload it can carry is 1g when it is powered by an external power supply. Hence, no evidence could support the assumption that the robot in reference [1] can carry any quadrotor.

Comment R2-6:

Overall, I have major concerns about the novelty and impact of this work. In the meantime, I think the experiments are done thoroughly and the figures are of high quality. The robot combines many functions, and it is worthy of publication in a field specific journal. Minor concerns:

In my view, the title is misleading as the authors claim their robot is capable of “collaborative flying”. However, when I review this paper, it looks like the robot is simply carrying a mini-quadrotor to go through a tunnel. I do not think this represents any flight capability. I suggest rephrasing the title.

Response to Comment R2-6:

Thank you for the comment. We have revised the title of the paper to “Multimodal locomotion ultra-thin soft robots capable of crawling, climbing, swimming and collaborating for exploration of narrow spaces”.

The collaboration was demonstrated by two Type-A TS-Robots assisting each other to perform transitions between horizontal and vertical surfaces in Fig.7C. Further, the collaboration was demonstrated by adding additional crawling locomotion capability to a drone, so the miniaturised drone is enabled to crawl through a narrow tunnel, which cannot be done by the drone itself (Movie 12). It may seem simple, but no other DEA-Driven robot has reported this capability of carrying other robots, and the reasons and challenges are explained at the end of “Response to Comment R2-5”.

[Amendments in the revised main manuscript on page 1 - Response to Comment R2-6]

Multimodal locomotion ultra-thin soft robots capable of crawling, climbing, swimming and collaborating for exploration of narrow spaces

Comment R2-7:

Minor concerns:

I think the authors shouldn't just highlight the few demonstrations in Table 1 without comparing other performance metrics, such as speed, efficiency, and cost of transport.

Response to Comment R2-7:

Thank you for the comment. We added a new figure to list the key performance metrics of the state-of-the-art DEA-driven soft robots and our robots, including robot thickness, speed, and payload/self-weight ratio.

[Amendments in the revised supplementary materials on page 52 - Response to Comment R2-7]

Robot's name [Reference number]
(thickness, ratio of payload/self-weight)

★ Robot locomotion facilitated by EA-Pad
★ Robot locomotion facilitated by directional friction foot

Fig. S29. Performance comparison of our robots with other DEA-actuated robots. (The speed data for the saddle-shaped robot was calculated based on its crawling demonstration in a confined space with a height of 10mm: Supplementary material-movie 6).

Response to Reviewer 3 (Amendments are marked in yellow)

General comments:

In this work, one kind of multimodal locomotion ultra-thin soft robots, consisting of dielectric elastomer actuators and electroadhesive pads, have been proposed, which is capable of crawling, climbing and swimming. Several experiments have been conducted to well demonstrate the multimodal locomotion ability and the potential application of exploring narrow spaces. This work can be recommended for publishing in Nature Communications after addressing following comments.

Response to general comments:

Thank you for the time reviewing our paper and for the insightful and constructive comments. To address your comments and concerns, we have added a substantial number of new experiments, data, and clarifications to improve the manuscript. In the following part of the document, we presented our response to addressing your comments point-by-point. The contents added to the manuscript and supplementary information is marked in yellow.

Specific comments:

Comment R3-1

In the section of Introduction, although achieving multimodal locomotion is important for narrow spaces, the main challenge to develop dielectric elastomer actuator based multimodal locomotion is unclear. Considering that many dielectric elastomer actuator based robots have been proposed, it is better to explain why multimodal locomotion is challenging.

Response to Comment R3-1:

Thank you for the constructive comment. The design and making of thin DEA-Driven robots for multimodal locomotion are challenging, especially *when needing to access near-millimetre and submillimetre-high narrow spaces*, which are the targeted application scenarios in this paper. While multi-modal locomotion in free/large spaces has been achieved by many existing robots, when it comes to transforming a thin soft robot into a multi-modal one, significant challenges need to be addressed:

1. Challenge 1 (Ch1): locomotion in multiple domains (e.g. solid/liquid media) in narrow spaces often requires a robot to perform different gaits (*ref-42*), linear gait for crawling on solid surfaces or through narrow gaps, and undulating gait for swimming in liquid. In contrast, a robot with a single gait may perform locomotion in one domain but may find it challenging in another (*ref-19*), or it may perform multiple gaits but need to be actuated by an external magnetic field, which could be challenging to be used in built environments (*ref-42*).

However, our thin robot, capable of 2 gaits, has been proven capable of crawling in narrow spaces and swimming as well as amphibious locomotion (i.e. transition between liquid and solid).

2. Challenge 2 (Ch2): Multimodal locomotion may need several robots to work together to transition between different terrains (*ref-43*), such as transitioning from horizontal to vertical surfaces and carrying a miniaturised drone to cross narrow gaps. This requires the robot to have a high payload/self-weight ratio and stiffness to carry objects, such as another TS-Robot or drone.

However, our concepts of thin soft robots offer the possibility to couple (via passive/active joints) with other similar robotic structures, i.e. form a cluster, which helps each other to pass through different domains of narrow spaces; this capability of the robot was enabled by high payload/self-weight ratio design (29 times self-weight).

We have added a discussion in the manuscript to clarify the challenges of multimodal locomotion.

[Amendments in the revised main manuscript on page 23 - Response to Comment R3-1]

The design and fabrication of thin DEA-driven robots for multimodal locomotion are challenging, especially when accessing near-millimetre and submillimetre-high narrow spaces, which are the targeted application scenarios in this paper. While multi-modal locomotion in free/large spaces has been achieved by many existing robots, when it comes to designing a thin soft robot for accessing narrow spaces in built environments and performing multimodal locomotion, significant challenges need to be addressed:

1. Challenge 1 (Ch1): locomotion in multiple domains (e.g. solid/liquid media) in narrow spaces often requires a robot with a thin thickness to perform multiple gaits (*42*), such as linear gait for crawling on solid surfaces or through narrow gaps and undulating gait for swimming in liquid. In contrast, a robot with a single gait may perform locomotion in one domain but may find it challenging in another (*19*), or it may perform multiple gaits but need to be actuated by an external magnetic field, which could be challenging to be used in built environments (*42*).
2. Challenge 2 (Ch2): Multimodal locomotion in narrow spaces may need several robots to work together to transition between different terrains (*43*), such as transitioning between horizontal and vertical surfaces and crossing large gaps. This requires the robot to have a high payload/self-weight ratio and stiffness to carry another robot to assist each other in crossing the barriers.

Comment R3-2

In Fig. 1, many potential applications of the robots have been illustrated. However, those applications are not verified by experiments. It is easy to cause confusion.

Response to Comment R3-2:

Thank you for your constructive comment. Please kindly consider that Fig. 1 intends to provide an overview of generic applications to inspire the readers academically. As such, we would like to clarify that in the scenarios of Fig. 1, the following generic barriers can usually be present in industrial applications, including but not limited to:

1. Narrow gaps on horizontal or vertical surfaces (“Security” scenario – Fig. 1: inspection under containers)
2. Complicated paths under narrow gaps (“Energy” scenario – Fig. 1: inspection underneath chemical storage tanks)
3. Hybrid solid-liquid environment (“Chemistry” scenario – Fig. 1: inspection transitioning in liquid-contaminated ground)
4. Transitions between horizontal and vertical surfaces (“Aerospace” scenario – Fig. 1: inspection of aerospace engines)
5. Transitioning between the above scenario via coupled thin soft robots (e.g., Fig. 1 C) and/or by using a “transporter”, (e.g. drone, Fig. 1 E) (“Energy” and “Security” scenario – Fig. 1: inspection underneath chemical storage tanks and containers, with the assist of a drone for delivering the TS-Robot next to the targets)

The proposed TS-Robots were tested for overcoming each type of these barriers separately in the experiment present in Fig. 4 (narrow gaps on horizontal or vertical surfaces), Fig. 5 (complicated paths under narrow gaps), Fig. 6 (hybrid solid-liquid environment), Fig. 7 (transitions between horizontal and vertical surfaces) and Fig. 8 (deep trenches/gaps between targeted surfaces).

We do not claim that our robots can be ready to be used as a high technology-readiness level product in all the applications in Fig. 1, but we intend to inspire the community with these possibilities. Further, we selected one real-world application from the aerospace industry to demonstrate the TS-Robots, which can be completed during the revision. Here, we used one TS-Robot to perform in-situ inspections of the magnetic field degradation of an electrical generator through the air gap between its rotator and stator. The demonstration details can be found in “Response to Comment 2-2”.

Comment R3-3

In the text S1 to S4, analytical models of the TS-DEAs have been proposed. I noticed that the viscoelasticity has been taken into consideration. However, Fig. S5 only shows the static experimental results and verification. Viscoelasticity usually leads to rate-dependent hysteresis effect that has serious influence on the dynamic response of DEA and the control strategy of the DEA-based robots. It is better to further analyze the viscoelasticity of DEAs.

Response to Comment R3-3:

Thank you for the comment. In the revised paper, we provided additional experimental data and verifications related to the hysteresis effect on the dynamic response of the proposed DEA.

Fig. S7. Model calibration and verification of the TS-DEAs. (A) Experimental force-displacement data of Type-A TS-DEA to fit the value of constant material property k ; (B) Nonlinearization of the constitutive behavior of tensioning mechanism material; (C) Type-A TS-DEA linear actuation model verification; (D) Type-A TS-DEA bending actuation model verification; (E) Type-B TS-DEA linear actuation model verification; (F) Displacement of Type-A TS-DEA as a function of time over 12 cycles; (G) Type-A TS-DEA displacement as a function of voltage for the last three cycles, with negligible creep effect.

Comment R3-4

The electroadhesive force of the electroadhesive pad usually relies on the material, surface of the substrate. The influence of voltage and substrate should be characterized.

Response to Comment R3-4:

Thank you for the constructive comment. We have characterised the 3D-printed electro-adhesive pad on different materials of the substrate. The results have been added to the Supplementary file Text S7.

[Amendments in the revised supplementary file on pages 23-24 - Response to Comment R3-4]

Text S7. EA-Pad characterisation

To characterize the tangential adhesion force of the EA-pads, a force measurement device was designed, as shown in Fig. S9. A linear motor was used to horizontally move the EA-pad on the substrate at a constant speed of 0.1 mm/s, while step voltages were applied during the process. The tangential force between the EA-pad and the substrate during motion, $F_{tangential}$, was measured by a force sensor connected to the wire.

Fig. S9 Experimental setup of tangential force measurement of EA-pad

The normal electrostatic adhesion force between the EA-pad and the substrate, F_{normal} , is proportional to the square of the applied voltage, while the tangential force is jointly determined by the friction coefficient between the EA-pad and the substrate and the normal force, and can be expressed as (25):

$$F_{tangential} = \mu_f F_{normal} = \mu_f k_E V_E^2 = \mu_E V_E^2 \quad (S46)$$

where F_{normal} is the normal electrostatic adhesion force, V_E is the voltage applied to the EA-pad, k_E is a dimensional constant determined by the geometric shapes and materials of the electrodes and the substrate, μ_f represents the equivalent friction coefficient between the electrostatic feet and the substrate, μ_E is the tangential adhesion constant, equal to $\mu_f k_E$. Since the breakdown voltage of the electrostatic feet ranges from 2600V to 2800V, we applied a set of voltage values ranging from 500V to 2500V with an incremental increase of 500V. The tangential forces measured on surfaces of paper, wood, PET, and PVC are shown in Fig. S10. Experimental results indicate that the tangential force between the EA-pad and the substrate is approximately proportional to the square of the voltage, consistent with the prediction of Eq. (S46). It is worth noting that when applying a voltage of 2500V, the maximum tangential forces exerted by the EA-pad on the surfaces of the four materials are 0.59N, 0.38N, 0.49N, and 0.53N respectively, which are 1204, 776, 1000, and 1082 times the weight of the EA-pad.

Fig. S10 Characterisation of the tangential force of the EA-pad. The red markers represent the measured values of the tangential force, the blue dashed line indicates the fitting curve, and the error bars represent the standard deviation of five measurements for each data point.

Comment R3-5

The control system is much larger and heavier than that of robots. It may lead to huge challenge for the application in narrow space.

Response to Comment R3-5:

Thank you for the constructive comment. In this paper, we aim to develop robots to explore narrow spaces (e.g., submillimetre and near-millimetre gaps) in high-value industrial equipment, such as aero-engines, aircraft, gas turbine generators and nuclear facilities. To access narrow spaces, we use small-diameter electrical power cables to actuate the TS-Robots, enabling the thin-thickness design of the robots and the control system to be located remotely (outside the narrow spaces). Hence, the size and weight of the control system will not be a hurdle to the proposed applications.

As an example, we have added a demonstration of using the TS-Robot to inspect electrical aero-engines to clarify the scope of the possible applications. The demonstration details can be found in “Response to Comment 2-2”.

Further, there are also engineering solutions to make the control system small. This system's enabling factor is the elastomer's thin thickness, allowing using a low voltage (220V) to drive the robot (*ref-19*). In the revision, we also demonstrated that the proposed TS-Robot can be driven by a low voltage (220V), to illustrate the possibility of using a small control system for the proposed system. The demonstration details can be found in “Response to Comment 2-2”.

REVIEWERS' COMMENTS

Reviewer #1 (Remarks to the Author):

I appreciate the efforts from the authors to address the comments from the reviewers, and in particular the effort to show robotic performance with novel materials. Given the improved status of the paper, in particular showing novel materials can improve robotic functionality and increase speed, I consider this article is now worthy of publication in Nature Communications.

Reviewer #2 (Remarks to the Author):

I appreciate the authors' extensive effort in preparing the revision. There has been a significant amount of data added to the paper, as well as insightful discussion to the applications. I think the paper can be accepted to Nature Communications. I have a few minor issues that I hope the authors can address.

First, the authors claim their DEA-driven robot is the fastest. I hope they can do a more thorough literature review. I am still under the impression that DEA-driven robot, especially tethered robot, can be much faster because of recent improvement in power and design. For instance, in [new 1], the authors show a 3D-printed DEA robot can achieve 4 BL/s, much faster than the one reported in this work. 3D printing trades off quality for fabrication scalability, so I believe DEAs can make very fast terrestrial robots. I hope the authors can comment on this paper and the prospect of fast DEA robots.

Second, I hope the study can be broad enough to consider similarly functioned microrobot beyond the DEA research community. In terms of low-profile (thin) robot, electrostatic thin film robot has shown similar capabilities inspecting thin gap, as shown in [new 2]. In that work, the authors have incorporated cameras on top of a very thin robot and use them to inspect gaps.

Regarding fast microrobots with small thickness, another work [new 3] has reported a very thin robot that is power autonomous and can move at over 20 cm/s or 7.5BL/s. In my view, the robots in [new 2 and 3] can be applied in the applications mentioned in this manuscript.

In summary, I appreciate the effort that the authors have put in, and I think the paper is worthy of publication based on the extensive amount of data shown in the work. I hope the discussion can be more comprehensive and general so it can have higher impact on the research community. My request is optional, but I hope the authors can comment on the 3 papers I listed below.

[New 1] Zhu, Y., Liu, N., Chen, Z., He, H., Wang, Z., Gu, Z., Chen, Y., Mao, J., Luo, Y. and He, Y., 2023. 3D-printed high-frequency dielectric elastomer actuator toward insect-scale ultrafast soft robot. ACS Materials Letters, 5(3), pp.704-714.

[New 2] Wang, H., York, P., Chen, Y., Russo, S., Ranzani, T., Walsh, C. and Wood, R.J., 2021. Biologically inspired electrostatic artificial muscles for insect-sized robots. *The International Journal of Robotics Research*, 40(6-7), pp.895-922.

[New 3] Liang, J., Wu, Y., Yim, J.K., Chen, H., Miao, Z., Liu, H., Liu, Y., Liu, Y., Wang, D., Qiu, W. and Shao, Z., 2021. Electrostatic footpads enable agile insect-scale soft robots with trajectory control. *Science Robotics*, 6(55), p.eabe7906.

Reviewer #3 (Remarks to the Author):

The authors have addressed most of my concerns. However, I still think that Fig. 1 should be used for illustrating the design of the robots, instead of the potential applications. Although those potential applications are presented, they are not achieved. For the journal like NC, it is better to focus on the the fundamental science problem.

Review Comments

Response to Reviewer 1

Comments R1-1:

I appreciate the efforts from the authors to address the comments from the reviewers, and in particular the effort to show robotic performance with novel materials. Given the improved status of the paper, in particular showing novel materials can improve robotic functionality and increase speed, I consider this article is now worthy of publication in Nature Communications.

Response to Comment R1-1:

Thank you for your insightful comments. They have significantly contributed to improving the quality of our paper.

Response to Reviewer 2 (Amendments are marked in blue)

Comments R2-1:

I appreciate the authors' extensive effort in preparing the revision. There has been a significant amount of data added to the paper, as well as insightful discussion to the applications. I think the paper can be accepted to Nature Communications. I have a few minor issues that I hope the authors can address.

First, the authors claim their DEA-driven robot is the fastest. I hope they can do a more thorough literature review. I am still under the impression that DEA-driven robot, especially tethered robot, can be much faster because of recent improvement in power and design. For instance, in [new 1], the authors show a 3D-printed DEA robot can achieve 4 BL/s, much faster than the one reported in this work. 3D printing trades off quality for fabrication scalability, so I believe DEAs can make very fast terrestrial robots. I hope the authors can comment on this paper and the prospect of fast DEA robots.

Second, I hope the study can be broad enough to consider similarly functioned microrobot beyond the DEA research community. In terms of low-profile (thin) robot, electrostatic thin film robot has shown similar capabilities inspecting thin gap, as shown in [new 2]. In that work, the authors have incorporated cameras on top of a very thin robot and use them to inspect gaps.

Regarding fast microrobots with small thickness, another work [new 3] has reported a very thin robot that is power autonomous and can move at over 20 cm/s or 7.5BL/s. In my view, the robots in [new 2 and 3] can be applied in the applications mentioned in this manuscript.

In summary, I appreciate the effort that the authors have put in, and I think the paper is worthy of publication based on the extensive amount of data shown in the work. I hope the discussion can be more comprehensive and general so it can have higher impact on the research community. My request is optional, but I hope the authors can comment on the 3 papers I listed below.

[New 1] Zhu, Y., Liu, N., Chen, Z., He, H., Wang, Z., Gu, Z., Chen, Y., Mao, J., Luo, Y. and He, Y., 2023. 3D-printed high-frequency dielectric elastomer actuator toward insect-scale ultrafast soft robot. ACS Materials Letters, 5(3), pp.704-714.

[New 2] Wang, H., York, P., Chen, Y., Russo, S., Ranzani, T., Walsh, C. and Wood, R.J., 2021. Biologically inspired electrostatic artificial muscles for insect-sized robots. *The International Journal of Robotics Research*, 40(6-7), pp.895-922.

[New 3] Liang, J., Wu, Y., Yim, J.K., Chen, H., Miao, Z., Liu, H., Liu, Y., Liu, Y., Wang, D., Qiu, W. and Shao, Z., 2021. Electrostatic footpads enable agile insect-scale soft robots with trajectory control. *Science Robotics*, 6(55), p.eabe7906.

Response to Comment R2-1:

Thank you for your instructive suggestions. We have included the papers provided by Reviewer 2 in the manuscript and have added corresponding discussions to enhance the quality of our work, aiming to make a better impact in the community. (Literature [new 1] has been added to Table 1 and Fig. S29 to compare with the existing DEA-driven robots; As [new 2] and [new 3] use different actuation methods, not DEA-driven, we have discussed their performance on page 24 in the discussion of the main manuscript)

[Amendments in the revised manuscript on page 24 - Response to Comment R2-1]

In the field of soft robotics, thin robots can also be achieved by different actuation methods, such as electrostatic¹³ and piezoelectric⁴⁵, which have been used to build insect-scale robots⁴⁵ for fast locomotion on flat surfaces. However, dielectric elastomer actuators (DEAs) offer significant advantages such as high flexibility, large strain, and high power density, which enables multi-modal locomotion by single or multiple robots.

[Amendments in the revised supplementary file on page 52 - Response to Comment R2-1]

Fig. S29. Performance comparison of our robots with other DEA-actuated robots. (The speed data for the saddle-shaped robot was calculated based on its crawling demonstration in a confined space with a height of 10mm: Supplementary material-movie 6).

Response to Reviewer 3 (Amendments are marked in yellow)

Comments R3-1:

The authors have addressed most of my concerns. However, I still think that Fig. 1 should be used for illustrating the design of the robots, instead of the potential applications. Although those potential applications are presented, they are not achieved. For the journal like NC, it is better to focus on the fundamental science problem.

Response to Comment R3-1:

Thank you for your insightful suggestions, we have moved Fig. 1 (showing the potential applications of the proposed ultra-thin robots) to the Discussion section in the manuscript.

[Amendments in the revised manuscript on page 23 - Response to Comment R3-1]

We envisage that such robots are promising solutions for several real-world applications, including inspection of confined spaces within industrial plants and civil engineering buildings, performing security surveys of confined spaces, as well as accessing and observing hazardous environments (Fig. 8).

Fig. 8 Concepts of the Thin-Soft Robots (TS-Robot) for potential applications. (a) Type-A TS-Robot (left) and Type-B TS-Robot (right), targeting for navigation in narrow gaps on horizontal or vertical surfaces (“Energy” scenario); (b) Asymmetric configuration of Type-A TS-Robot, targeting for the hybrid solid-liquid environment (“Chemistry” scenario); (c) Twin Type-A TS-Robot, targeting for transitions between horizontal and vertical surfaces (“Security” scenario); (d) Serial Kinematics TS-Robot, targeting for transitions between horizontal and vertical surfaces (“Aerospace” scenario); (e) One example of TS-Robots collaborating with other types of robots – Flying-TS-Robot, targeting for Transitioning between surfaces by using a carrier (“Security” and “Energy” scenario).